# Allosteric inhibition of CFTR gating by CFTRinh-172 binding in the pore

Xiaolong Gao [1] ✉, Han-I Yeh [1,2,3], Zhengrong Yang[4], Chen Fan [5,6], Fan Jiang[4], Rebecca J. Howard [5,6], Erik Lindahl [5,6], John C. Kappes[4,7] & Tzyh-Chang Hwang [1,2,3] ✉

Loss-of-function mutations of the CFTR gene cause the life-shortening genetic disease cystic fibrosis (CF), whereas overactivity of CFTR may lead to secretory diarrhea and polycystic kidney disease. While effective drugs targeting the CFTR protein have been developed for the treatment of CF, little progress has been made for diseases caused by hyper-activated CFTR. Here, we solve the cryo-EM structure of CFTR in complex with CFTRinh-172 (Inh-172), a CFTR gating inhibitor with promising potency and efficacy. We find that Inh-172 binds inside the pore of CFTR, interacting with amino acid residues from transmembrane segments (TMs) 1, 6, 8, 9, and 12 through mostly hydrophobic interactions and a salt bridge. Substitution of these residues lowers the apparent affinity of Inh-172. The inhibitor-bound structure reveals re-orientations of the extracellular segment of TMs 1, 8, and 12, supporting an allosteric modulation mechanism involving post-binding conformational changes. This allosteric inhibitory mechanism readily explains our observations that pig CFTR, which preserves all the amino acid residues involved in Inh-172 binding, exhibits a much-reduced sensitivity to Inh-172 and that the apparent affinity of Inh-172 is altered by the CF drug ivacaftor (i.e., VX-770) which enhances CFTR's activity through binding to a site also comprising TM8.

In the human body, the chloride channel Cystic Fibrosis Transmembrane conductance Regulator (CFTR) mediates chloride transport across the cell membrane in multiple epithelial-lining organs, including intestines, biliary ducts, pancreas ducts, respiratory airways, and sweat ducts[1–4]. This trans-epithelial movement of chloride through CFTR plays a critical role in the secretion or absorption of salt and water in these organ systems; malfunction of CFTR channels causes human diseases. Specifically, loss-of-function mutations of the CFTR gene are the root cause of cystic fibrosis (CF)[5]; conversely, numerous studies have linked upregulated CFTR activity to secretory diarrhea and

polycystic kidney disease, resulting from excessive salt/water transport into the intestinal lumen and kidney cysts, respectively[6–9].

Topological analysis of CFTR places it in the ATP-binding Cassette (ABC) transporter family, which harnesses the energy from ATP hydrolysis to move substrates across cell membranes[10–12]. Structurally, ABC proteins possess two transmembrane domains (TMDs) forming the substrate translocation pathway and two cytosolic nucleotide-binding domains (NBDs) that recruit and hydrolyze ATP as the energy source. While many ABC proteins retain two catalysis-competent NBDs, only one of the ATP binding sites (i.e., site 2) in CFTR can

[1]Dalton Cardiovascular Research Center, University of Missouri-Columbia, Columbia, MO 65211, USA. [2]Institute of Pharmacology, National Yang Ming Chiao Tung University, College of Medicine, Taipei, Taiwan. [3]Membrane Protein Structural Biology Research Center, National Yang Ming Chiao Tung University, Taipei, Taiwan. [4]Heersink School of Medicine, University of Alabama School of Medicine, Birmingham, AL 35233, USA. [5]Department of Applied Physics, Science for Life Laboratory, KTH Royal Institute of Technology, Solna, Sweden. [6]Department of Biochemistry and Biophysics, Science for Life Laboratory, Stockholm University, Solna, Sweden. [7]Research Service, Birmingham Veterans Affairs Medical Center, Veterans Health Administration, Birmingham, AL 35233, USA. ✉e-mail: xgdz2@missouri.edu; hwangt@health.missouri.edu

hydrolyze ATP, with site 1 being hydrolysis-incompetent[13–16]. Unique to CFTR is a regulatory domain (R domain) that sits between the two TMD/NBD complexes, hence preventing ATP-binding induced NBD dimerization. Activation of CFTR is achieved by spontaneous dislodgement of the R domain from the inhibitory position, followed by protein kinase A (PKA)-dependent phosphorylation of multiple serine residues in the R domain that prevents the R domain from returning to the inhibitory position[17–21]. Once the R domain is phosphorylated, ATP-binding induced NBD dimerization and subsequent hydrolysis of ATP in site 2 are coupled to the opening and closing of the CFTR pore, respectively[12,22,23].

Decades of research on CFTR biochemistry, physiology, and pharmacology have led to the successful development of reagents that directly target the dysfunctional CFTR protein in CF patients[24–29]. Mechanisms of such CFTR modulators, namely CFTR correctors and potentiators, have also been extensively studied[30–45]. On the other hand, the possibility of using small molecules to inhibit hyperactivated CFTR in diseases like secretory diarrhea and polycystic kidney disease has not been as rigorously exploited[46–49]. To date, two types of CFTR inhibitors have been reported: one that blocks the ion-conducting pathway and is hence named CFTR blockers (e.g., glibenclamide[50,51] and GlyH-101[52]), and the other that inhibits CFTR gating (e.g., Inh-172[47]). Although our previous study suggests that Inh-172 inhibits CFTR gating by inducing conformational changes that obstruct the pore[53], where Inh-172 binds and what conformational changes take place following Inh-172 binding are not known.

Here, we report a cryo-EM structure of CFTR in complex with Inh-172. Structure and molecular dynamics (MD) simulations support stable binding of the drug in the pore through hydrophobic interactions and a salt bridge between the carboxylate in Inh-172 and the side

chain of lysine 95. Mutagenesis and electrophysiological studies show that substitution of residues involved in the binding decreases the apparent affinity of Inh-172. Compared to the control structure without Inh-172, this inhibitor-bound structure shows major re-orientations of the extracellular parts of three transmembrane segments (TMs 1, 8, and 12), supporting an allosteric modulation mechanism involving binding-induced conformation changes in the TMDs. This allosteric mechanism underlying the gating modulation by Inh-172 is further corroborated by studies with pig CFTR which preserves all the residues for binding and with the CFTR potentiator VX-770 that decreases the inhibitory potency of Inh-172.

## Results
### Determination of the structure of E1371Q-CFTR
Previous cryo-EM studies of both the human[54,55] and zebrafish CFTR[20,21] utilized the BacMam system for the protein expression[56]. In the current study, an engineered CFTR construct in a lentiviral vector developed by Yang et al.[57] was used for our cryo-EM studies. This expression system produces a robust yield of thermally stable CFTR proteins in both HEK-293 and Chinese hamster ovary (CHO) cells[57] (see Methods for more details of the construct), and the engineered CFTR protein so expressed is fully functional[58]. To verify that the CFTR proteins we produced maintain all the structural properties reported before[55], we first determined the molecular structure of phosphorylated E1371Q-CFTR in the presence of ATP because the additional E-to-Q mutation–an effective strategy used previously–abolishes ATP hydrolysis at CFTR's site 2 and thus locks the channel into a uniform conformation with a stable NBD dimer[23,59].

As shown in Fig. 1a, the yield of the E1371Q-CFTR mutant is high, ensuring that the sample can be effectively concentrated for cryo-grids

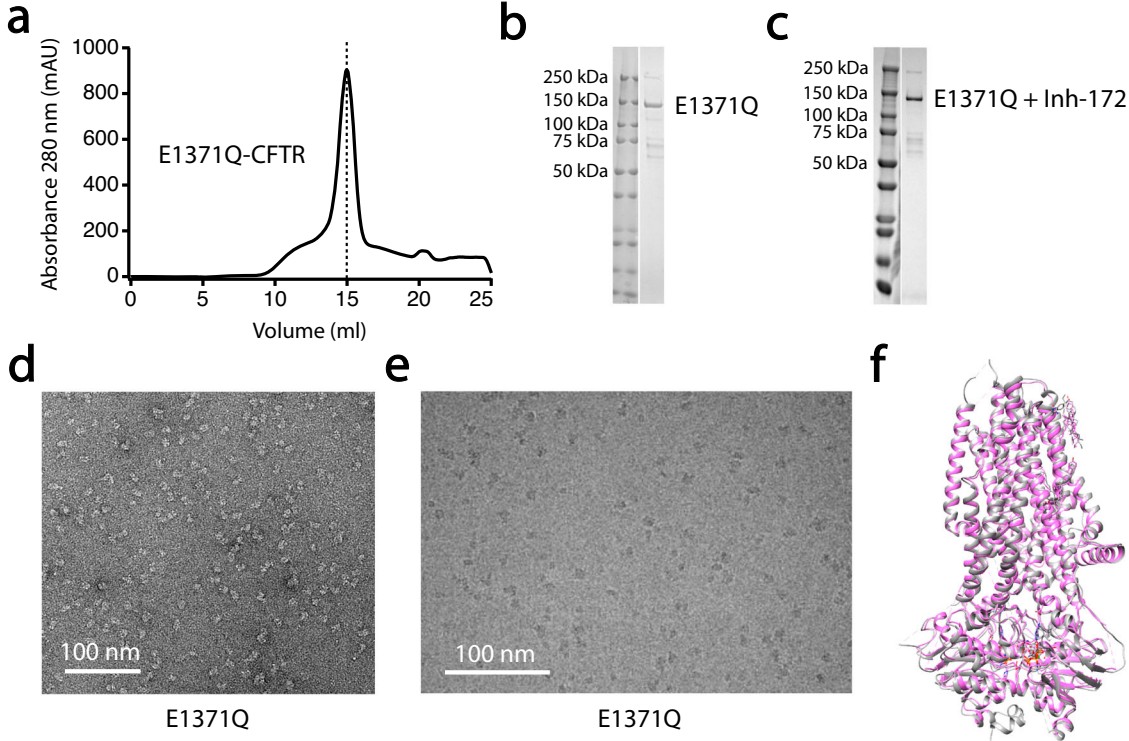

**Fig. 1 | Quality assessment of purified CFTR proteins for cryo-EM structural determination. a** Elution of E1371Q-CFTR in detergents on Superose 6 Increase column at 15 ml position. **b**, **c** SDS-PAGE gels showing the purity of CFTR for E1371Q and E1371Q plus Inh-172, respectively. The band positions correspond to the right molecular weight of mature deglycosylated CFTR; $n = 10$ for both experiments. Uncropped gels are provided as a Source Data file. **d** Negative staining image of

E1371Q-CFTR demonstrating notable mono-dispersity on the carbon film. Note the correct size of particles indicated by the scale; $n = 4$ for the experiment. **e** Cryo-image of E1371Q-CFTR proteins deposited on cryo-grids depicting similar mono-dispersity as seen in (**d**). **f** Overlay of the E1371Q-CFTR structure determined in the current work (pink) with the previously published E1371Q-CFTR (gray, PDB code: 6MSM) showing virtually identical molecular composition and overall conformation.

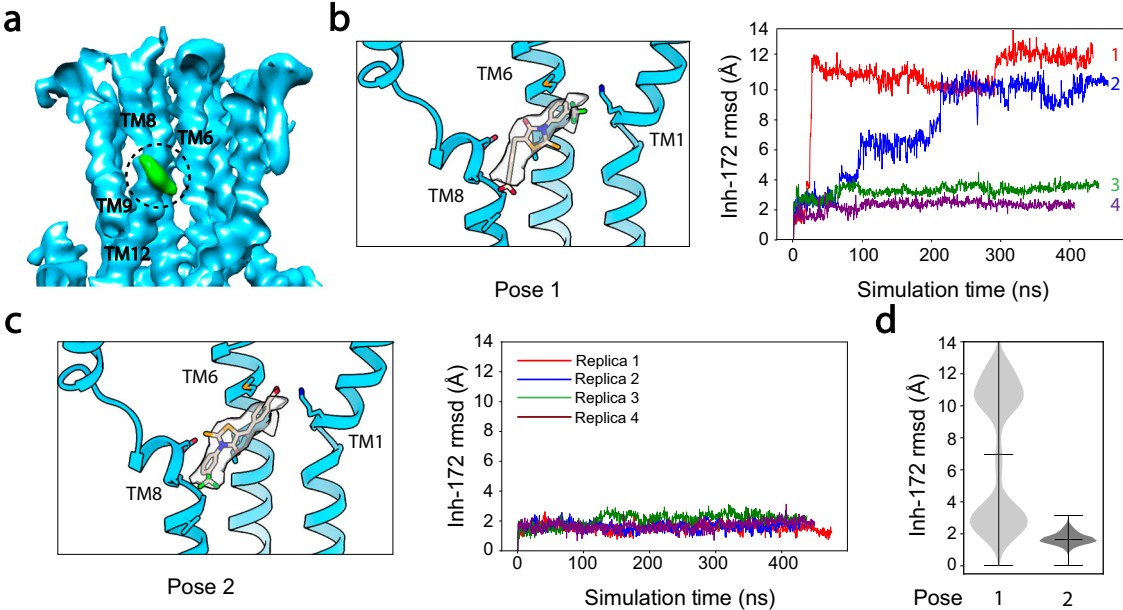

**Fig. 2 | Localization of the Inh-172 binding site on CFTR and MD simulations to determine drug orientation. a** Cross section of the cryo-EM map of E1371Q-CFTR/Inh-172 complex showing density corresponding to Inh-172 (green density in dashed circle) in the pore of CFTR. Surrounding TMs that comprise the binding site are labeled. Note that the map was low pass filtered and that TM1 (another TM contributing to the drug binding site) and parts of TM6 and TM8 were removed for a better view of the drug density. **b, c** MD simulations of two possible drug orientations. Lefts: the structure of Inh-172 can be fitted into the drug density in two poses. Only TMs 1, 6, and 8 are shown. Rights: Inh-172 RMSDs in its respective poses, calculated from MD simulations. Four replicates (colored separately) were performed for each pose. A comparison of Inh-172 dynamics at the binding pocket indicates that pose 2 in (**c**) is more stable in CFTR's pore. **d** Violin plot summarizing Inh-172 RMSDs during simulations in each pose.

freezing and that abundant protein particles can be imaged for cryo-EM structure determination. The sharp and symmetric elution peak of E1371Q-CFTR proteins from size exclusion chromatography (SEC) indicates that the sample has high purity and homogeneity, both of which are characteristics ideal for cryo-EM studies. The protein is eluted at ~ 15 ml position on the Superose 6 Increase column and the SDS-PAGE gel indicates a molecular weight (MW) of ~ 150 kDa of the eluted protein, which is the right MW for our CFTR construct (Fig. 1b). When E1371Q-CFTR was copurified with Inh-172, the compound was added during both cell culture and protein purification (see Methods). Because of the small MW of Inh-172, the position of the band for the E1371Q-CFTR/Inh-172 complex was not altered on the SDS-PAGE gel (Fig. 1c).

Before the purified proteins were subjected to single-particle cryo-EM studies, we evaluated their dispersity with negative staining under a transmission electron microscope. The negative staining images of E1371Q-CFTR, in which individual intact CFTR protein molecules can be readily discerned, show that the proteins are mono-dispersed and not prone to aggregate at room temperature (Fig. 1d). Upon deposition on cryo-grids, the mono-dispersity of E1371Q-CFTR protein particles is maintained (Fig. 1e), which facilitates later particle picking and processing. With this high-quality sample, the electron density map of E1371Q-CFTR was determined at an overall resolution of 3.4 Å where densities of the side chains for most amino acids are clearly defined (Fig. S1). Overlay of our molecular model of E1371Q-CFTR with previously published structure[55] shows negligible differences between the two structures with an RMSD of 1.1 Å for all paired Cα (Fig. 1f). Thus, the mutations introduced to stabilize the CFTR protein (see Methods) do not disturb the overall structure of the E1371Q-CFTR protein.

### Identification of an Inh-172 binding site in CFTR's ion permeation pathway
Using the above phosphorylated, ATP-bound E1371Q-CFTR, we next determined the cryo-EM structure of the E1371Q-CFTR/Inh-172 complex to an overall resolution of 3.6 Å (Fig. S2). A comparison of the Inh-172 free and Inh-172 bound electron density maps revealed an extra density, located inside the pore of CFTR (circled green density in Fig. 2a), matching the size of Inh-172. The density sits right below the narrowest region, a place long conjectured as CFTR's gate and selectivity filter[60–63]. When we fitted the structure of Inh-172 into the identified density, we found that the density could accommodate Inh-172 assuming either of the two binding poses: one with its carboxylate facing TM8 (pose 1, Fig. 2b) and the other facing TM1 (pose 2, Fig. 2c). We thus turned to MD simulations to determine which binding pose is energetically more favorable. When the compound was built into its density with its trifluoromethyl group pointing towards the nearby M348 in TM6 and K95 in TM1 (Fig. 2b, left), highly fluctuating dynamics were observed in two out of four MD simulations, each over 400 ns (Fig. 2b, right, red and blue). In one replicate, Inh-172 moved dramatically away from its density even within the first 50 ns, as indicated by the RMSD of Inh-172 relative to its original position (red curve in Fig. 2b, right), suggesting an unstable binding of Inh-172 in this pose. In contrast, a consistent, more stable interaction (<3 Å RMSD in all replicate simulations) was seen in pose 2, in which the trifluoromethyl group of Inh-172 interacts with TM8 (Fig. 2c) and TM12. Figure 2d summarizes MD simulation results supporting that binding pose 2 represents a more likely orientation of Inh-172 in the binding pocket.

Based on pose 2, MD simulations further revealed that Inh-172 interacts with its site particularly through hydrophobic and salt bridge interactions (Fig. 3a). The hydrophobic trifluoromethyl phenyl head of Inh-172 is wedged into a crevice formed mainly by TMs 8, 9, and 12, involving hydrophobic side chains including V920, A923 and L927 from TM8, I1000 and A1104 from TM9, and I1139 from TM12 (Fig. 3a–c and Fig. S3). In addition, the central thiazolidine ring of Inh-172 makes contact with three aromatic amino acids F310, F311, and W1145. At the other end of the compound, the 4-carboxyphenyl ring of Inh-172 is coordinated by the side chain of M348 from TM6 through a hydrophobic interaction and the indole ring of W1145 from TM12 through a π stacking interaction.

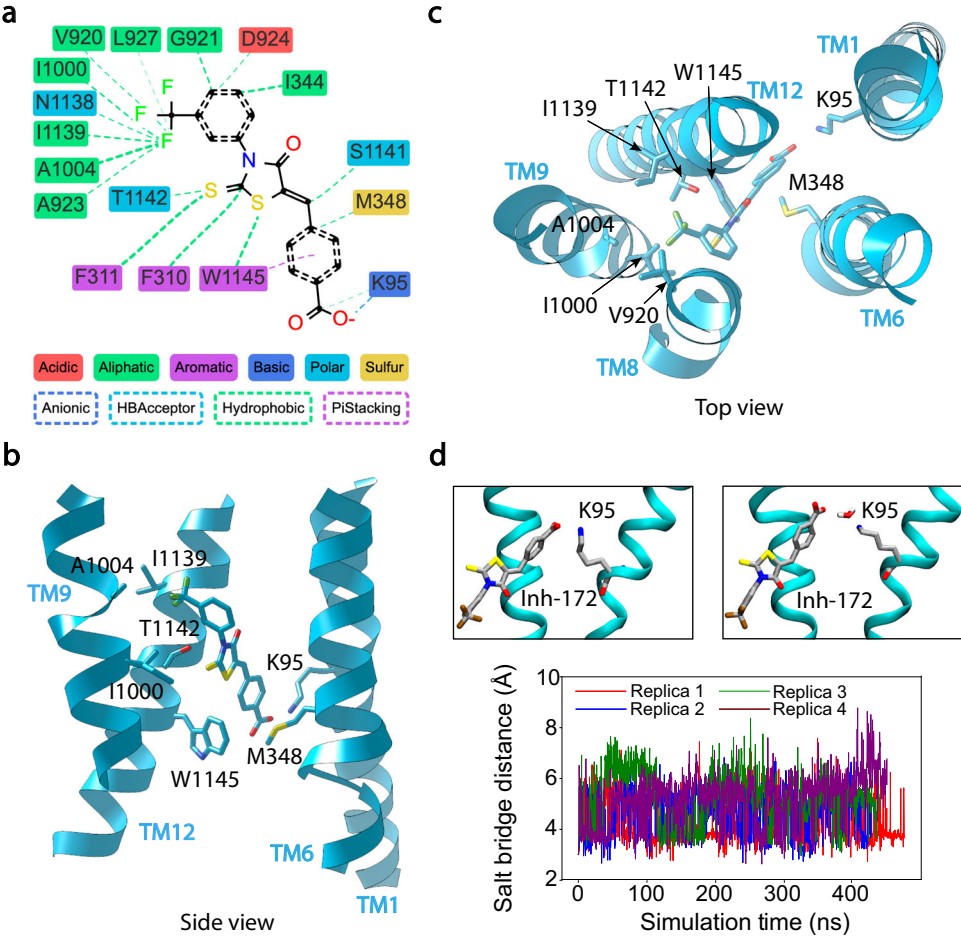

**Fig. 3 | Interactions of Inh-172 with amino acid residues constructing its binding site. a** Stabilization of Inh-172 through hydrophobic interactions and a salt bridge. Types of interaction are specified between each amino acid and the inhibitor. **b, c** Three-dimensional views of the inhibitor binding at its binding pocket. Key residues involved in the binding of Inh-172 from each TM are labeled. TM8 was removed for better visualization in (**b**). **d** Direct or water-mediated salt bridge between the Inh-172 and K95 during MD simulations. Top: A zoom-in view of the MD simulation with direct contact (left, ~3.5 Å) and water-mediated (right, ~ 5.5 Å) salt bridge between the carboxylate of Inh-172 and the ammonium group of K95. Bottom: MD simulations measure the distance of the salt bridge, which remains relatively constant over the entire simulation time. Four repeats of the simulation were performed as indicated by different colors.

In addition to the hydrophobic and aromatic interactions described above, pose 2 also enables a salt bridge between the carboxylate in Inh-172 and the ammonium group of K95 in TM1 (Fig. 3b–d). During all four replicate MD simulations in this pose, this charge-pair distance (measured between the nitrogen zeta of K95 and the center of mass of Inh-172's carboxylate oxygen atoms) fluctuated around 3.5 Å and 5.5 Å, corresponding to direct contact or water-mediated salt bridge (Fig. 3d). The importance of this salt bridge for Inh-172 binding was further validated by our electrophysiological experiments, in which 0.5 μM Inh-172 inhibited K95A-CFTR current by only ~40%, while 80% WT-CFTR current was inhibited by the same concentration of Inh-172 (Fig. 4a, b). Quantitative dose-response measurements show that the K95A mutation increases the $IC_{50}$ of Inh-172 by 10-fold (from 0.08 ± 0.01 μM to 0.85 ± 0.05 μM), presumably due to the breaking of this salt bridge (Fig. 4e, Table 1). Similarly, other binding site mutants tested in this work, including M348K-, T1142A-, T1142I-, W1145A-, W1145L- and W1145Y-CFTR, all demonstrate various degrees of increased $IC_{50}$ for Inh-172 (i.e., rightward shift of the dose-response curves, Fig. 4 and S4, Table 1), corroborating their roles in the binding of Inh-172. Of note, all electrophysiological experiments on the binding site mutations were carried out in the wild-type background for more straightforward data interpretation, rather than in the E1371Q background, as this mutation dramatically increases the potency of Inh-172 via a state-dependent effect on binding proposed previously[53].

## TMD2 plays a key role in the inhibition of CFTR gating by Inh-172

We next superimposed the structures of phosphorylated ATP-bound E1371Q-CFTR with and without Inh-172 to look for Inh-172-induced conformational changes proposed previously[53]. The measured global RMSD between these two structures is 1.5 Å for all paired Cα with a smaller RMSD of 1.1 Å for all paired Cα in their NBDs, indicating the existence of some but limited global conformational changes upon Inh-172 binding to the E1371Q-CFTR and the preservation of a canonical NBD dimer in the Inh-172 bound state (Fig. 5a). Comparing the positions of the residues directly involved in the binding of Inh-172 (Fig. 3a) between drug-free and drug-bound structures, we observed negligible changes of their positions with an RMSD of 1.1 Å (Fig. 5b). Since the identified binding site is located immediately below the narrowest region of the pore, the bulky Inh-172 molecule could in theory inhibit CFTR by directly obstructing the chloride permeation pathway just like other CFTR channel blockers[50,52,64]. However, we found Inh-172-induced conformational changes in the extracellular mouth of the channel. When the extracellular parts of CFTR's two TMDs were aligned for the Inh-172 bound and Inh-172 free structures (Fig. 5c), it became clear that in the drug-bound structure, the extracellular segment of TM12 (from V1129 to L1143) wedges into the central axis for 30°, while extracellular segments of TM1 (from V93 to Y109) and TM8 (from T910 to L926) tilt away from the central axis for 10° and 13°, respectively, resulting in a closed pore conformation. The calculated

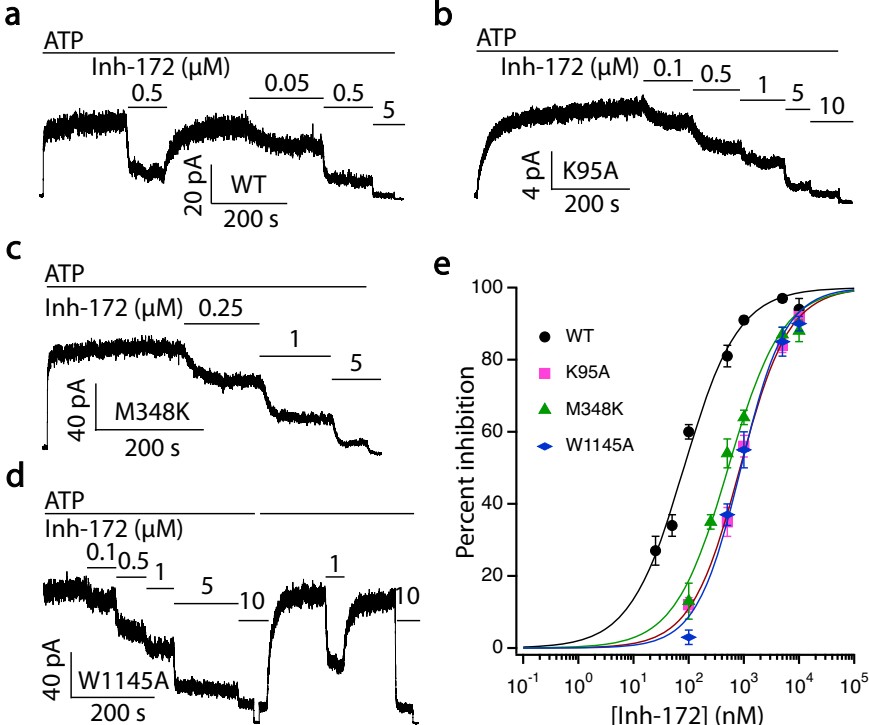

**Fig. 4 | Mutations at the Inh-172 binding site decrease the potency of Inh-172 on CFTR. a–d** Electrophysiological recordings from excised inside-out membrane patches with a holding voltage of -30 mV to measure the currents of WT-, K95A-, M348K- and W1145A-CFTR inhibited by Inh-172 at marked concentrations in the presence of 2 mM ATP. Inward currents are displayed as upward deflections in all figures. **e** Dose-response curves of Inh-172 for WT-, K95A-, M348K-, and W1145A-

CFTR. All three mutations cause a rightward shift of the dose-response relationship. Data were fitted with Eq. (2) as described in Methods. The $IC_{50}$ and Hill coefficient values are reported in Table 1. $n$ = 8, 6, 8, and 5 cells for WT-, K95A-, M348K-, and W1145A-CFTR respectively. Data represent mean ± SEM. Source data are provided as a Source Data file.

RMSD for these three segments between the two structures is 4.3 Å. Interestingly, the position of TM8 in the CFTR/Inh172 complex of the current study resembles, but is not identical to, the position of TM8 in the unphosphorylated, ATP-free CFTR structure where the NBDs are

### Table 1 | $IC_{50}$s of Inh-172 and Hill coefficient values for all CFTR constructs

| Constructs Values | $IC_{50}$ (μM) | Hill Coefficient |
|---|---|---|
| Human WT | 0.08 ± 0.01 | 0.88 ± 0.09 |
| K95A | 0.85 ± 0.05 | 0.98 ± 0.05 |
| M348K | 0.52 ± 0.06 | 0.90 ± 0.10 |
| T1142A | 0.27 ± 0.01 | 0.81 ± 0.02 |
| T1142I | 1.49 ± 0.07 | 1.12 ± 0.05 |
| W1145A | 0.88 ± 0.09 | 1.08 ± 0.12 |
| W1145L | 0.64 ± 0.06 | 0.82 ± 0.07 |
| W1145Y | 0.87 ± 0.03 | 0.92 ± 0.03 |
| Pig WT | 1.86 ± 0.12 | 0.83 ± 0.04 |
| h-pTMD1-h | 0.11 ± 0.01 | 0.98 ± 0.10 |
| p-hTMD1-p | 2.21 ± 0.07 | 1.03 ± 0.03 |
| h-pTMD2-h | 4.02 ± 0.28 | 0.69 ± 0.04 |
| p-hTMD2-p | 0.06 ± 0.02 | 0.54 ± 0.08 |
| h-pTM8-h | 0.64 ± 0.02 | 0.80 ± 0.02 |
| p-hTM8-p | 0.72 ± 0.09 | 0.84 ± 0.10 |
| Human WT + 200 nM VX-770 | 0.25 ± 0.02 | 1.07 ± 0.10 |
| Human WT + 1 μM VX-770 | 0.42 ± 0.01 | 0.98 ± 0.02 |

separated[54]. Such structural differences suggest that the Inh-172-induced closed state is a closed conformation that is different from the closed state determined before in ref. 54. The same observations were also made by Young et al.[65].

The unique rearrangements of TM1, TM8, and TM12 in the inhibitor-bound structure distant to Inh172's binding site suggest an allosteric inhibitory mechanism involving post-binding conformational changes. An allosteric modulation mechanism dictates that the sensitivity of the channel to Inh-172 is determined by the binding equilibrium as well as by the equilibrium of the conformational change reaction following binding. It has long been established that CFTR orthologs from different species show different sensitivities to Inh-172[66,67], but the mechanism behind such difference remains unknown. Taking the identified binding site into consideration, we were puzzled by the fact that pig CFTR has a much-reduced sensitivity to Inh-172 despite its 100% conservation of those residues involved in Inh-172 binding and ~92% overall sequence homology to human CFTR (Fig. S5). As shown in Fig. 6a, f, pig WT-CFTR has a ~ 20-fold higher $IC_{50}$ of Inh-172 than human WT-CFTR (1.86 ± 0.12 μM vs. 0.08 ± 0.01 μM, Table 1). Interestingly, when the TMD2 of human CFTR was replaced with pig CFTR's TMD2 (pTMD2), the dose-response relationship of Inh-172 for the chimera channel (h-pTMD2-h-CFTR) drastically shifts to the right (Fig. 6b, f, green curve vs. black curve; Table 1), supporting the idea that the TMD2 outside the Inh-172 binding site also plays a role in determining the apparent affinity of Inh-172. Furthermore, when the TMD2 of pig CFTR was replaced with human CFTR's TMD2, the resulting chimera channel (p-hTMD2-p-CFTR) assumes a similar Inh-172 sensitivity to human CFTR (Fig. 6c and f, blue curve vs. black curve; Table 1). In contrast, swapping TMD1 between these two orthologs did not alter their respective apparent affinity (Fig. 6d–f, red curve vs. black curve and orange curve vs. purple curve; Table 1). As the amino acid sequence of the Inh-172

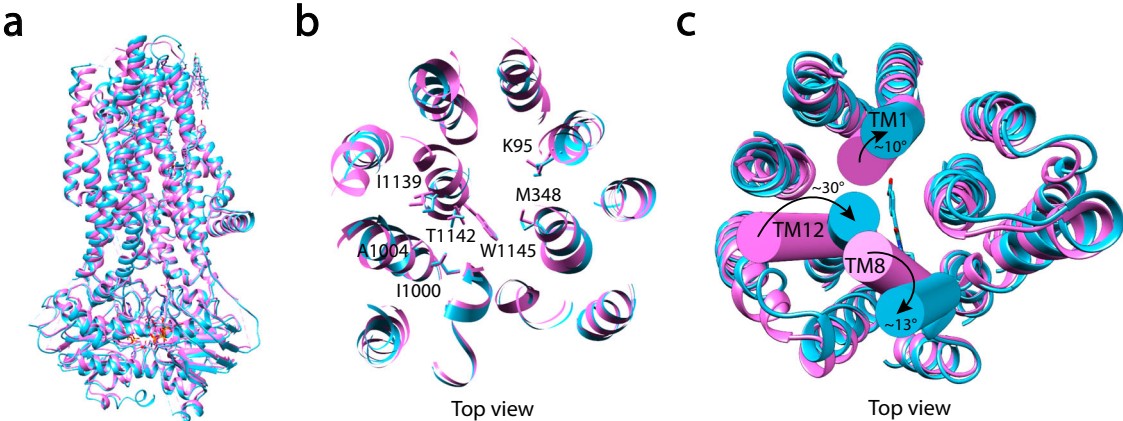

**Fig. 5 | Inh-172 induces conformational changes in the extracellular parts of TMs 1, 8, and 12. a** Superimposition of E1371Q-CFTR (pink) and Inh-172 bound E1371Q-CFTR (cyan) reveals limited global conformational changes between drug-free and drug-bound structures. **b** Comparison of the positions of key residues involved in Inh-172 binding in drug-free and drug-bound E1371Q-CFTR from panel (**a**). For a clearer view, Inh-172 was removed from the drug-bound structure and the structures were represented as flat ribbons to highlight the relevant residues. **c** Inh-172 induced movements of the extracellular segment of TMs 1, 8, and 12. Angles are measured with the cylinder presentations. Note minimal change in the orientations of other TMs.

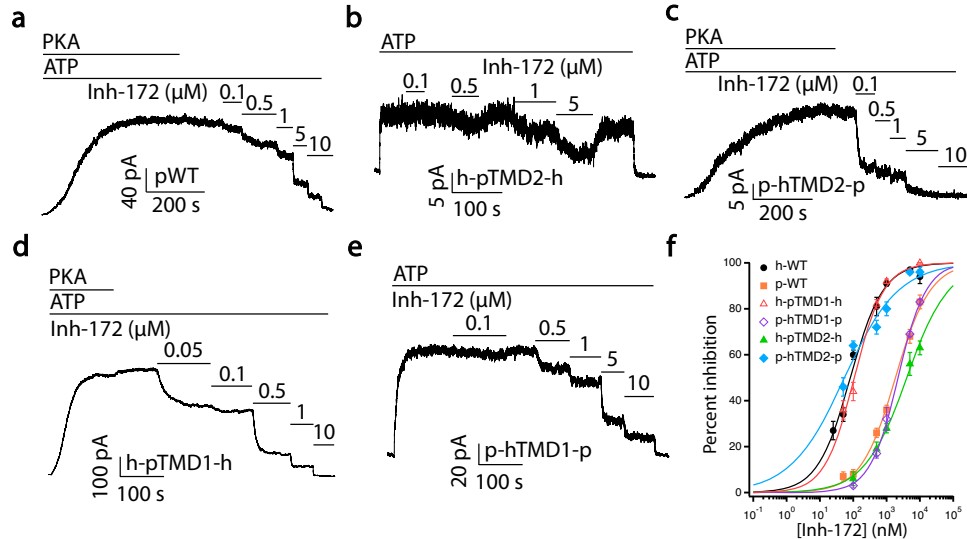

**Fig. 6 | Inh-172 allosterically modulates CFTR's activity through TMD2.**
**a** A representative recording of pig WT-CFTR (pWT-CFTR) currents in an inside-out patch showing inhibition by various concentrations of Inh-172 (cf. Fig. 4a).
**b** Responses of the h-pTMD2-h-CFTR chimera to different concentrations of Inh-172 showing that replacing the TMD2 of human CFTR with the TMD2 of pig CFTR drastically reduces the sensitivity of the channel to Inh-172. **c** The chimera p-hTMD2-p-CFTR exhibits much greater sensitivity to Inh-172 compared with pWT-CFTR in (**a**). **d, e** Current inhibition by Inh-172 on h-pTMD1-h-CFTR (human CFTR with pig CFTR's TMD1, **d**) and p-hTMD1-p-CFTR (pig CFTR with human CFTR's TMD1, **e**). **f** Dose-response curves of hWT-, pWT-, h-pTMD1-h-, p-hTMD1-p-, h-pTMD2-h- and p-hTMD2-p-CFTR. Data were fitted similarly as in Fig. 4e, and results are summarized in Table 1. $n = 8, 11, 4, 3, 4$, and 7 cells for hWT-, pWT-, h-pTMD1-h-, p-hTMD1-p-, h-pTMD2-h-, and p-hTMD2-p-CFTR, respectively. Data represent mean ± SEM. Source data are provided as a Source Data file.

binding site is the same for both human and pig CFTR, these results suggest that conformational changes of TMD2 outside Inh-172's binding site contribute to the overall inhibition of CFTR currents by Inh-172. Interestingly, just changing three amino acids in TM8 of pig CFTR (highlighted in yellow, Fig. S5) to those of human CFTR shifted the dose-response curve toward that of human CFTR (Fig. 7a, c). Similarly, human CFTR with those three amino acids replaced with pig's sequence has a dose-response relationship closer to that of pig CFTR (Fig. 7b, c). As the motion of TM8 is critical to CFTR's gating[21], and mutations at this region alter the potency of Inh-172 (Fig. 7), we next tested the effect of the CFTR potentiator VX-770, which, contrary to the Inh-172, binds on the external side of the TM8 kink[32], on the sensitivity of WT-CFTR to Inh-172. If the binding of Inh-172 in the pore solely accounts for the inhibitory effects of Inh-172, the increased open probability with a prolonged open time by VX-770[30] is expected to shift the dose-

response relationship of Inh-172 to the left[53]. Contrary to this prediction, the presence of VX-770 shifts the dose-response relationship of Inh-172 to the right (Fig. 8). This result can be readily explained by the idea that the CFTR potentiator VX-770 and inhibitor Inh-172 compete for TM8 for activated or inactivated state, respectively, as they bind to the opposite side of TM8 (see Discussion for more details).

## Discussion
The current study reported two cryo-EM structures of E1371Q-CFTR proteins produced by a lentiviral expression system. The E1371Q-CFTR proteins so produced show highly uniform conformations owing to the E-to-Q mutation abolishing ATP hydrolysis at CFTR's site 2[22,68]. Our cryo-EM analysis of the E1371Q-CFTR mutant reached an overall resolution of 3.4 Å and revealed a conformation closely resembling the previous structure obtained using the BacMam system[55], affirming the compatibility of the

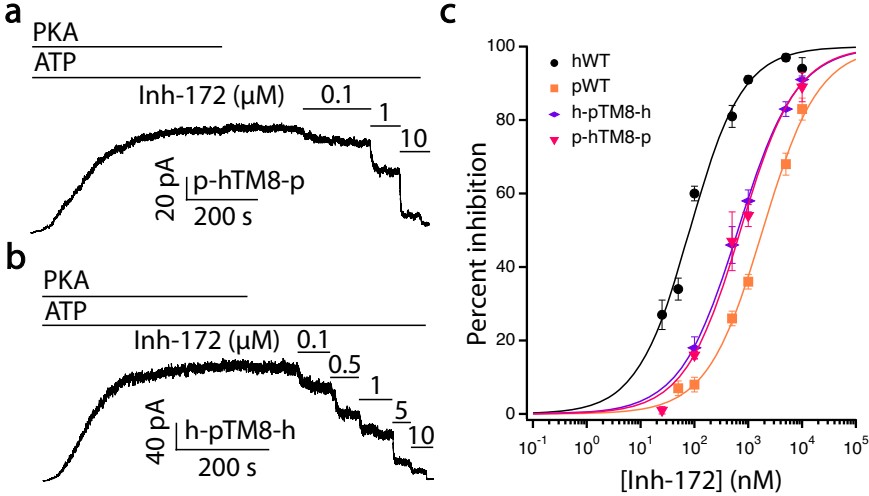

**Fig. 7 | Effects of TM8 in determining the sensitivity of CFTR to Inh-172.**
**a** A representative electrophysiological recording showing inhibition induced by Inh-172 on a pig CFTR construct where three amino acids in TM8 (G926, L930 and L932, marked in yellow in Fig. S5) were replaced by their corresponding counterpart in human CFTR's TM8. **b** A recording from an inside-out patch showing inhibition induced by Inh-172 on a human CFTR construct where three amino acids in TM8 (T925, M929 and F931, marked in yellow in Fig. S5) were replaced with their counterpart in pig CFTR's TM8. Of note, these three amino acid residues are located at the hinge region of TM8 where the helix breaks. **c** Dose-response curves for h-pTM8-h-CFTR and p-hTM8-p-CFTR. Human WT-CFTR and pig WT-CFTR curves from Fig. 6 are replotted for comparison. Data were fitted as in Fig. 4e and fitting results are summarized in Table 1. $n = 8$, 11, 6, and 5 cells for hWT-, pWT-, h-pTM8-h-, and p-hTM8-p-CFTR, respectively. Data represent mean ± SEM. Source data are provided as a Source Data file.

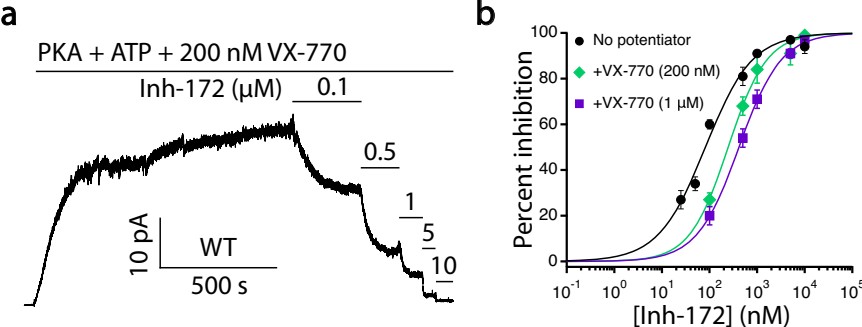

**Fig. 8 | The apparent affinity of Inh-172 is lowered in the presence of VX-770.**
**a** A representative recording of WT-CFTR in response to various concentrations of Inh-172 in the continuous presence of 200 nM VX-770. **b** Dose-response relationships of Inh-172 on WT-CFTR in the absence and presence of 200 nM or 1 μM VX-770. The $IC_{50}$s of Inh-172 are: 0.08 ± 0.01 μM without VX-770, 0.25 ± 0.02 μM with 200 nM VX-770 and 0.42 ± 0.01 μM with 1 μM VX-770 (also see Table 1). $n = 8$, 4, and 5 cells for no potentiator, 200 nM VX-770, and 1 μM VX-770, respectively. Data represent mean ± SEM. Source data are provided as a Source Data file.

lentiviral expression system with the BacMam system in producing high-quality CFTR proteins for cryo-EM studies. Moreover, the Inh-172 binding site identified in our work is virtually identical to that reported by Young et al. using the BacMam vector[65]. Mutagenesis studies at the binding site also support this common finding as both studies show that K95A and T1142I mutations lower the potency of Inh-172. Of note, the E1371Q-CFTR construct was used for structural determination in both reports, and similar conformational changes at the extracellular region of TMDs were observed upon binding of Inh-172: the extracellular segments of TM1 and TM8 move away from the pore and the extracellular segment of TM12 tilts into the pore. While Young et al. propose binding of Inh-172 alters the ATP turnover rate in the NBDs, our data support an allosteric modulation nature of Inh-172's action on CFTR most likely via conformational changes in the TMD2.

Binding of Inh-172 in the CFTR pore was first proposed by Caci et al. based on the observation that mutations at R347, a residue in the pore-forming TM6, lower the apparent affinity of Inh-172[69]. However, R347 does not contribute to the identified binding site in the structures of the CFTR/Inh-172 complex. One possible explanation for Inh-172's lowered apparent affinity observed by Caci et al. with mutations

at R347 is that by breaking the pore-stabilizing salt bridges of R347-D924 and R347-D993[70,71], those mutations alter the pore's structural integrity and hence affect the binding of Inh-172 and/or the subsequent Inh-172 induced conformational changes, a possibility also raised by Young et al.[65].

Our previous functional studies suggested that Inh-172, unlike classical pore blockers[50–52], works as a gating inhibitor[53]: conformational changes following Inh-172 binding lead to an "inactivated" state of CFTR. The same study also implicated a state-dependent binding of Inh-172 by showing that the apparent affinity of Inh-172 is increased when the open state is stabilized. More specifically, maneuvers (e.g., E1371S mutation or pyrophosphate) that stabilize the NBD dimer and lock open the channel increase the apparent affinity of CFTR for Inh-172. Thus, using a similar mutant CFTR with an even longer open time (i.e., E1371Q in this work) is an ideal strategy to identify the binding site for Inh-172 and the conformation of the "inactivated" state. Indeed, the structure of E1371Q-CFTR with Inh-172 bound provided the answer for both inquiries (Figs. 2 and 5).

Since the identified Inh-172 binding site is located in the pore, in theory, the binding of bulky Inh-172 could inhibit CFTR current by

simply obstructing the pore[72], but our cryo-EM structure of E1371Q-CFTR/Inh-172 complex also reveals important conformational differences between inhibitor-bound and inhibitor-free states (Fig. 5c). Compared to the inhibitor-free structure, the one with inhibitor bound shows significant movements of the extracellular ends of TMs 8 and 12, and a minor shift of the extracellular end of TM1. These structural changes induced by Inh-172 binding have several implications. First, the observation supports the proposition that Inh-172 acts through an allosteric modulation mechanism[53]. Second, the new orientations of both TM8 and TM12 in the Inh-172 bound structure are not the same as what was observed in the unphosphorylated, ATP-free conformation[54,65]—hence the CFTR conformation with this newly arranged TMs 8 and 12 is a closed state different to what has been reported before. However, during normal ATP-dependent gating, this state is seldom visited and/or rather unstable in the absence of Inh-172; otherwise, one should have observed frequent long-lasting closed events in single-channel recordings of WT-CFTR. Third, the fact that the extracellular portions of TM8 and TM12 can assume at least three different positions indicates that these two segments are flexible and mobile, a required property for serving as an important gating apparatus for CFTR. A similar idea was previously proposed[21].

If we accept the idea that the modus operandi of Inh-172 is allosteric in nature, the classic mechanism modulating ion channel (or enzyme) function predicts a tighter binding of Inh-172 to the "inactivated" state than to the open state[73,74]. Previous functional studies indeed showed an extremely stable "inactivated" state of Inh-172 bound E1371S-CFTR[53], and the current structural data also indicate the predominance of an Inh-172 bound structure distinct from the Inh-172 free conformation. One would expect to see structural differences in the binding site between Inh-172 bound and Inh-172 free E1371Q-CFTR structure as each should represent the inactivated state and the open state, respectively. We were thus surprised to find that those amino acid residues involved in coordinating Inh-172 binding were not visibly altered when we aligned their positions in E1371Q-CFTR structures with and without Inh-172 (Fig. 5b). Therefore, we caution our readers that the proposed allosteric inhibition mechanism for Inh-172 is based on the observation that binding of Inh-172 induces conformational changes of TM1, 8 and 12 that are distant to the binding site (hence the term allosteric). Since no significant structural changes were observed within the binding site upon Inh-172 binding, our allosteric inhibition mechanism diverges from the classical allosteric model, where conformational changes within the modulator binding site typically follow modulator binding.

Although the exact reason is unknown for the absence of Inh-172-induced conformational changes within the binding site, and more studies are needed to address this fundamental issue, here we offer three possibilities. First, our structural analysis may not reveal the presumed fine differences in binding interactions because of insufficient local resolution that could limit the accuracy of amino-acid side chain assignment at the binding site in both maps. Second, Inh-172 may indeed act through a process different from the classical allosteric modulation mechanism. Third, the structure of the ATP-bound E1371Q-CFTR is not an open state. Whereas the first two possibilities are speculative and can be verified with structures of higher resolution available in the future, the third is supported by the fact that none of the structures solved so far under the E1371Q background show a patent pore; their ion permeation pathways are all obstructed at the narrowest segment of the pore[21,32,33,45,55]. This latter observation is puzzling by itself and remains to be addressed, but it has been suspected that the cryo-EM experimental conditions including the use of detergent for solubilization of plasma membrane may affect the conformation of the channel[21,75].

Despite the above-mentioned caveat, the structure of phosphorylated, ATP-bound E1371Q-CFTR does represent an "activated" state close to the open channel conformation: a flickery closed state buried in the opening burst[21] or a quasi-open state coined by Simon and Csanady[76], respectively. In contrast to these structural insights of a quasi-open state, the conformation of the closed state during ATP-dependent gating of phosphorylated WT-CFTR is far from clear. Structural and functional studies suggest the existence of many possible post-hydrolytic closed states with various degrees of NBD separation[77–82]. At this juncture, the only structure approximating this more physiologically relevant closed state is the unphosphorylated, ATP-free WT-CFTR with separated NBDs (PDB code: 5UAK)[54]. By measuring the distances between the amino acid residues involved in Inh-172 binding to estimate the dimension of the binding site, we found that the binding site is more compact (i.e., the distances are shorter) in Inh-172 bound structure than in the unphosphorylated, ATP-free structure (Fig. S6). Assuming that Inh-172 binds to the same location when NBDs are not in a dimeric configuration, we reasoned that Inh-172 should bind more tightly to the site when the NBDs are dimerized. In other words, this comparison suggests that the binding site is not as fit when the two NBDs are separated—hence the binding of Inh-172 is not as tight, a supposition supported by Kopeikin et al., (2010) showing the closed WT-CFTR channel (in the absence of ATP) exhibits a lower affinity to Inh-172. Also consistent with this interpretation is that the recovery from inhibition upon washout of Inh-172 occurs more slowly in the presence of ATP than in the absence of ATP[53], presumably because dimerized NBDs and closer TMDs maintain a tighter binding site for Inh-172.

Besides state-dependent binding, another practical implication of the allosteric modulation mechanism is that the measured affinity of the ligand is determined by not only the association/dissociation of the compound but also the reaction of conformational changes following binding[73]. This principle should apply to both allosteric gating potentiators[34] and inhibitors. Thus, our measured $IC_{50}$ of Inh-172 will be affected by both the $K_d$ of the pure binding reaction that Inh-172 binds to the channel ($O \leftrightharpoons O_{Inh-172}$), and the equilibrium constant of the reaction for conformational changes from Inh-172 bound open state to the inactivated state ($O_{Inh-172} \leftrightharpoons I_{Inh-172}$). In other words, the easier for the channel to sojourn to the "inactivated" state after Inh-172 binding, the lower the concentration of Inh-172 to reach the same degree of inhibition—that is, a leftward shift of the dose-response curve (and vice versa). This gating modulation mechanism, however, predicts two distinct closed states in the presence of Inh-172 in single-channel recordings. One possible reason for missing such observation is that if the reaction rate of channel inactivation (i.e., $O_{Inh-172}$ to $I_{Inh-172}$) is higher than the dissociation rate of Inh-172 (i.e., $O_{Inh-172}$ to O), electrophysiological recording may not be able to capture the block/unblock events ($O \leftrightharpoons O_{Inh-172}$) because upon Inh-172 binding the channels immediately become inactivated.

While future studies using compounds with a similar action as Inh-172 may substantiate our proposed inhibitory mechanism, the principle of allostery may explain why the $IC_{50}$ of Inh-172 for pig CFTR is drastically shifted to the right (Fig. 6, Table 1) despite the 100% conservation of those amino acid residues involved in Inh-172 binding (highlighted in green, Fig. S5). Moreover, because the major conformational changes occur in TM8 and TM12 (two TMs in TMD2) with Inh-172 binding, it is also unsurprising that transplanting pig TMD2 to human CFTR created a construct with a much-reduced sensitivity to Inh-172 (and vice versa). Although our data won't be able to allow us to pinpoint the exact underlying cause of such sensitivity change, the observation that switching three amino acids in TM8 between human CFTR and pig CFTR can cause apparent affinity change for Inh-172 reinforces the importance of TM8 in CFTR's gating. It is possible that different side chains at the kink of TM8 introduce different dynamics for this region, especially given that methionine and phenylalanine (i.e., M929 and F931 in human CFTR's TM8) are more likely to participate in specific sidechain chemistry (e.g., methylthio- and π-orbital interactions, respectively) than leucine (L929 and L931 in pig CFTR's

TM8) which is basically hydrophobic. Our analysis here provides a conceptual framework for future studies of the mechanism underlying the action of Inh-172 in different species.

One final subject worth discussing is to ask how the binding of Inh-172 inside the pore promotes the motions of the TMs. The fact that the phosphorylated, ATP-bound E1371Q-CFTR conformation is well conserved regardless of the presence of various CFTR potentiators and correctors suggests that this is a stable conformation representing an activated CFTR channel. As discussed above, the extracellular segments of TM8 and TM12 in TMD2 can assume different positions and thus are more mobile. We hence propose a chemico-mechanical mechanism for the action of Inh-172: using the relatively immobile TMD1 (especially TM1 and TM6) as an anchor, Inh-172 drives the mobile TM8 and TM12 from the activated channel conformation along an energetically favorable trajectory to form the inactivated channel conformation with an obstructed pore. This hypothesis has two immediate implications: First, Inh-172 works not only because of the functional groups that can specifically interact with residues in TMD1 and TMD2, but the length of the compound is also critical so that the two TMDs at the inhibitor binding site can be properly bridged. Noticeably, Sonawane and Verkman carried out a structure-activity relationship study with 58 Inh-172 analogs and found the most potent CFTR inhibitor requires a trifluoromethyl group at position 3 of the phenyl ring and a 4-carboxyphenyl ring at the other end of the molecule[83]. These results are consistent with our cryo-EM structure that shows a salt bridge between the carboxylate of Inh-172 and lysine 95 in TMD1 and the hydrophobic interactions between the trifluoromethyl group and TM8/TM12 in TMD2. Second, as the observed lateral motions of TM8 and TM12 constitute the Inh-172-induced conformational changes that lead to the "inactivated" state, reagents that promote the activated state—that is, compounds that bind more tightly to the activated channel conformation—may counteract the action of Inh-172. Indeed, in Fig. 8, the presence of VX-770 reduces the potency of Inh-172. As VX-770 and Inh-172 bind to the opposite side of TM8 almost in direct juxtaposition, the two drugs become each other's antagonist for CFTR's gating. In other words, they are like playing a tug of war by steering TM8 to the position representing the activated or inactivated state of CFTR.

In conclusion, the current study identified the binding site of Inh-172 in the inner vestibule of CFTR's pore and the accompanied conformational changes in the extracellular segments of TMs 1, 8, and 12. This structural result is consistent with an allosteric modulation mechanism for Inh-172. A chemico-mechanical mechanism is proposed to guide future structure-based drug design targeting the identified Inh-172 binding site.

## Methods

### Cell culture and protein purification

The lentiviral vector for mammalian cell expression of stabilized human CFTR variants was modified with His10-SUMO* and 901FLAG affinity purification tags, and fused C-terminally with the enhanced green fluorescent protein (EGFP)[57]. Our construct also includes an amino acid deletion from F405 to L436 with six other mutations (M470V, S492P, S495P, A534P, I539T, and R555K) to improve the thermal stability of the protein, on top of the E1371Q mutation studied in this work. The Chinese Hamster Ovary (CHO) cells expressing CFTR were solubilized in Buffer W (20 mM HEPES, 150 mM NaCl, 10% glycerol, 0.2 mM TCEP, pH 7.5) containing 0.5% dodecyl-β-D-maltoside, 0.1% cholesteryl hemisuccinate (CHS), 2 mM DTT and Roche cOmplete™ EDTA-free Protease Inhibitor Cocktail (Millipore-Sigma, St. Louis, MO). The proteins were purified via a C-terminal STREP II tag using Strep-tactin resin (IBA LifeSciences, Göttingen, DE). The solubilizing detergents were replaced by 0.06% (w/v) digitonin via extensive washing while the protein was bound to the resin. The tagged protein was eluted in Buffer W containing 0.06% (w/v) digitonin and 4 mM *d*-

desthiobiotin. The eluted protein was concentrated to ~1 mg/ml and flash frozen. Following thawing, the purification tags were removed by incubating the protein with TEV protease on ice for 16 h at a roughly 10:1 mass ratio. The protein was phosphorylated and deglycosylated using PKA and PNGase F, respectively, during tag removal. The resulting mixture was subjected to size exclusion chromatography (SEC) on a Superose 6 Increase 10/300 GL (Cytiva, Marlborough, MA) gel-filtration column in Cryo-EM buffer (20 mM Tris, 200 mM NaCl, 0.06% digitonin, 2 mM DTT, pH 7.5) containing 0.2 mM MgATP. The CFTR peak center fractions were pooled, concentrated, and subjected to another round of SEC using the same conditions. Peak center fractions were again collected and supplemented with 2 mM MgATP before they were concentrated for cryo-EM grids preparation. The preparation of samples containing Inh-172 was the same as above, except that 10 μM Inh-172 was added into the cell culture media 2 hours before harvesting, and the same concentration of the compound was also included in all purification buffers. Additionally, the concentrated sample was spiked with 100 μM of Inh-172 after protein concentration.

### Cryo-EM grids freezing and data collection

The E1371Q-CFTR proteins collected from the SEC were concentrated to 6 mg/ml, and additional 6 mM ATP was added. Before freezing, 3 mM fluorinated Fos-Choline 8 was added to the sample for better particle distribution. 3.5 μL sample was deposited to each Quantifoil Au 1.2/1.3 400 mesh grid (Quantifoil, Großlöbichau, Germany) which was frozen using a Vitrobot Mark IV (Thermo Fisher Scientific, Waltham, MA) with the following conditions: 22 °C temperature, 100% humidity, 15 seconds wait time, 3 seconds blot time and 0 blot force. Data collection for the E1371Q-CFTR sample was carried out on a 300 kV Krios microscope (Thermo Fisher Scientific) equipped with a K3 camera (Gatan, Pleasanton, CA) running in the super-resolution mode using a pixel size of 0.529 Å. Each movie contains 40 frames with a total exposure time of 2 seconds and a total dose of 53.36 e⁻/Å². Defocus range was −0.5 μm to −2.5 μm.

The Cryo-EM grids preparation for the E1371Q-CFTR sample with Inh-172 (E1371Q/Inh-172) is similar to that for E1371Q proteins except that the E1371Q plus Inh-172 sample was concentrated to 4 mg/ml. During data collection, K3 camera was also operated under the super-resolution mode with a pixel size of 0.535 Å. Each movie contains 50 frames with a total exposure time of 2 seconds and a total dose of 65.69 e⁻/Å². Defocus range was between -1 μm and -1.8 μm. Leginon[84] and a 20 eV energy filter were used for both datasets collection.

### Cryo-EM data processing

For the E1371Q dataset, after manual examination of all exposures, 10159 movies were imported into Relion 4.0[85] for processing. Super-resolution micrographs were downscaled by a binning factor of 2 during motion correction. CTF estimation was carried out using CTFFIND-4.1[86]. Laplacian-of-Gaussian method was first used to pick particles for template generation before reference-based particle picking was performed on all images. Downscaled particles were initially extracted for faster multiple rounds of 2D classification to remove junk particles, which eventually resulted in 550,929 particles that were re-extracted without downscaling. A few more rounds of 2D classification were performed on these particles to further remove low-quality particles before 3D initial models were generated. Subsequently, two rounds of 3D classification were performed and a subset of 261,806 particles was refined into a map which was subjected to multiple rounds of CTF refinement and Bayesian polishing (Fig. S1). The overall resolution of the final map reached 3.4 Å (FSC$_{0.143}$).

Processing of the E1371Q/Inh-172 dataset with Relion is similar to that of the E1371Q dataset. Motion correction was done with a binning factor of 2 for 10700 manually curated super-resolution movies. CTF estimation was done with CTFFIND-4.1. After multiple rounds of 2D classification, 314,964 particles were subjected to 2 rounds of 3D

classification without image alignment. Eventually, a group of 140,837 particles was refined into a map showing the drug density at a resolution of 4.1 Å, which improved to 3.6 Å (FSC$_{0.143}$) after multiple rounds of CTF refinement and Bayesian polishing (Fig. S2).

## Model building

To build the E1371Q-CFTR model in this work, previously published model (PDB code: 6MSM)[55] was docked into our E1371Q map and manually adjusted according to our electron density map in Coot[87]. Deletion and point mutations (Δ(F405-L436) /M470V/S492P/S495P/ A534P/I539T/R555K) were introduced at the same time based on the sequence differences between our construct and the previously published one. Iterative refinement in PHENIX (1.20.1-4487)[88] and manual adjustment in Coot are performed for the model. The final model consists of the following amino acid residues: 11-383, 389-400, 438-635, 847-885, 908-1121, 1129-1171, 1207-1431.

To build the model of E1371Q plus Inh-172, our own E1371Q model and the structure of Inh-172 were fitted into the electron density map of E1371Q plus Inh-172. Restraints of Inh-172 were generated using the Grade Web Server (https://grade.globalphasing.org/cgi-bin/grade2_server.cgi) using its SMILES string from PubChem. Multiple rounds of refinement in PHENIX and manual adjustment in Coot were also performed. The final model consists of the following amino acid residues: 3-404, 437-637, 846-885, 915-1120, 1129-1173, 1203-1440. All map and model details are summarized in Table S1.

## Molecular dynamics simulations

Cryo-EM structures of CFTR with different Inh-172 poses were used as starting models for MD simulations. For the regions not built in the structure, we connected short gaps (<20 residues) with loops by SWISS-MODEL Server[89] while leaving longer gaps disconnected. As a result, the structure was split into three chains (chain A contains residues 3-637; chain B contains residues 846-885; chain C contains residues 915-1440). The simulation systems were set up in CHARMM-GUI[90]. The protein was embedded in a lipid mixture mimicking a mammalian cell membrane, with the outer leaflet containing 168 POPC, 20 POPE, and 80 cholesterol molecules and the inner leaflet containing 130 POPC, 50 POPE, 10 POPS, and 70 cholesterol molecules. The protein-lipid complex was subsequently solvated with TIP3P water, neutralized, and set to 150 mM with sodium and chloride ions. The Amber FF19SB[91] and GAFF2 force field[92] were used to describe the protein and Inh-172.

Simulations were performed using GROMACS 2021.5[93] at a temperature of 300 K using the velocity-rescaling thermostat[94] and Parrinello–Rahman barostat[95]. The LINCS algorithm was used to constrain hydrogen-bond lengths[96], and the particle mesh Ewald method[97] was used to calculate long-range electrostatic interactions. The systems were energy minimized and then equilibrated in three sequential steps, with gradual release of position restraints on heavy atoms for 8 ns, backbone atoms for 8 ns, and Cα atoms for 16 ns. Unrestrained production runs were simulated in four independent replicates, each for >400 ns.

For analysis, the MD simulation trajectories were aligned on the Cα atoms of the pore residues (78-108, 332-376, 969-1110, 1129-1168) using MDAnalysis[98]. Root mean square deviations (RMSDs) and salt-bridge distances of Inh-172 were calculated in VMD[99]. Small molecule interactions with the protein were quantified using ProLIF 1.1.0[100].

## Electrophysiology

CHO cells were grown at 37 °C in Dulbecco's modified Eagle's medium supplemented with 10% (vol/vol) fetal bovine serum (Sigma-Aldrich, St. Louis, MO). pcDNA plasmids carrying the desired CFTR construct were transfected to CHO cells using PolyFect transfection reagent (Qiagen, Hilden, Germany) together with green fluorescence protein (GFP) cDNA (pEGFP-C3, Takara Bio, Shiga, Japan) that helps identify expression of CFTR. Cells were first incubated with the transfection reagents at 37 °C

for 6-12 hours on sterile glass chips in 35-mm tissue culture dishes before transferred to 27 °C for incubation for 2–7 days. Primers for mutations were ordered from Integrated DNA Technologies (Coralville, IA) and were summarized in Supplementary Data. All CFTR mutants were prepared with QuikChange XL kit (Agilent, Santa Clara, CA) and sequenced at the DNA Core facility at University of Missouri-Columbia.

Patch-clamp pipettes were made from borosilicate capillary glass using a two-stage vertical puller (PP-81; Narishige, Amityville, NY). The pipette tips were fire polished with a homemade microforge to yield a pipette resistance of 2–5 MΩ when filled with a pipette solution containing (in mM): 140 NMDG-Cl (N-methyl-D-glucamine-chloride), 5 CaCl$_2$, 2 MgCl$_2$, and 10 HEPES, pH 7.4. For recordings, transfected cells on glass chips were transferred into a continuously perfused chamber on the stage of an inverted microscope (IX51; Olympus, Tokyo, Japan) with the perfusing solution containing (in mM): 150 NMDG-Cl, 2 MgCl$_2$, 10 EGTA, 8 Tris, and 10 HEPES, pH 7.4. The membrane patch was excised to an inside-out configuration with a seal resistance >40 GΩ. CFTR currents were recorded with an amplifier (EPC9; HEKA, Holliston, MA), filtered through an eight-pole Bessel filter (LPF-8; Warner Instruments, Hamden, CT) at 100 Hz and digitized to a computer at a sampling rate of 500 Hz with Pulse software (version 8.53; HEKA). The membrane potential was held at −30 mV. Solution exchange was achieved by a fast solution change system (SF-77B; Warner Instruments). Inh-172 was obtained from the CFTR Chemical Compound Program supported by the Cystic Fibrosis Foundation. All experiments were performed at room temperature.

The steady state mean currents were measured in the Igor Pro software (version 8.0; WaveMetrics). The mean baseline currents were subtracted from the steady state mean currents before the data were used to determine the degrees of inhibition. The degrees of inhibition by CFTR inhibitor 172 were calculated based on the equation:

$$\% \text{ inhibition} = \left(1 - \frac{I_{\text{Inh}-172}}{I_0}\right) \text{X } 100\% \qquad (1)$$

where $I_0$ and $I_{\text{inh}-172}$ represent uninhibited and inhibited currents, respectively. The IC$_{50}$ and Hill coefficients were obtained by fitting percent inhibition as a function of [Inh-172] with the Hill equation:

$$\% \text{ inhibition} = \text{base} + (\text{max} - \text{base}) / \left[1 + \left(\frac{\text{xhalf}}{x}\right)^{\text{rate}}\right] \qquad (2)$$

where max and base represent the minimum and maximum percent inhibition by Inh-172, $x$ represents the concentration of Inh-172, and the rate is the Hill coefficient. The estimated IC$_{50}$ (xhalf) and Hill coefficients are listed in Table 1 as mean ± standard deviation. Each patch-clamp recording was made from a separate cell.

## Figure preparation

All figure panels are prepared with Matplotlib[101], ProLIF (1.1.0), Igor Pro (version 8.0; WaveMetrics), and Chimera[102] before integrated in Illustrator (Adobe).

## Reporting summary

Further information on research design is available in the Nature Portfolio Reporting Summary linked to this article.

## Data availability

The cryo-EM maps and models were deposited into Electron Microscopy Data Bank (EMD-43011 for E1371Q and EMD-43014 for Inh-172 bound E1371Q) and Protein Data Bank (8V7Z for E1371Q and 8V81 for Inh-172 bound E1371Q), respectively. The raw micrographs for both datasets were deposited into Electron Microscopy Public Image Archive (EMPIAR-12042 for E1371Q and EMPIAR-12044 for Inh-172 bound E1371Q). MD simulation trajectory, parameter files are available

in Zenodo (https://doi.org/10.5281/zenodo.10407153). Source data are provided with this paper.

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

## Acknowledgements

The authors would like to acknowledge Min Su and Rowan Fink from the University of Missouri Electron Microscopy Core for their assistance in grids preparation. Some of this work was performed at the National Center for CryoEM Access and Training (NCCAT) and the Simons Electron Microscopy Center located at the New York Structural Biology Center, supported by the NIH Common Fund Transformative High Resolution Cryo-Electron Microscopy program (U24 GM129539,) and by grants from the Simons Foundation (SF349247) and NY State Assembly. We thank their staff Ed Eng, Eugene Chua, Huihui Kuang, and Kashyap Maruthi for their help with grid screening and data collection. We thank Shenghui Hu and Cindy Chu for their technical assistance in the construction of CFTR mutants and some electrophysiology experiments. The cluster used to analyze the Cryo-EM data is sponsored by NSF grant OAC-1919789. MD simulations were performed using the computing facilities of Swedish National Infrastructure for Computing (SNIC 2022/3-40) and supported by BioExcel (EuroHPC grant no. 101093290). This work is supported by fundings from Stockholm University (grant FV-5.1.2-0523-19 to C.F.), from the Swedish Research Council and Swedish e-Science Research Center (2019-02433, 2021-05806 to R.J.H. and E.L.), from NIH (R01DK055835 to T.H.), from NHRI (NHRI-EX112-11236SI to T.H.) and from Cystic Fibrosis Foundation grants (HWANG22G0 to T.H. and X.G.; KAPPES21XX0 to J.C.K.).

## Author contributions

Conceptualization: X.G. and T.H. Molecular biology, cell culture, and protein purification: Z.Y., F.J. and J.C.K. Cryo-EM data collection, analysis, model building and visualization: X.G. Electrophysiology: H.Y. MD simulation: C.F., R.J.H. and E.L. Funding acquisitions: X.G., E.L., J.C.K., and T.H. Writing: X.G., H.Y. and T.H. wrote the manuscript with inputs from all other authors.

## Competing interests

The authors declare no competing interests.
