## [Peer Review File · Nature Communications]

Allosteric inhibition of CFTR gating by CFTRinh-172 binding in the poreREVIEWER COMMENTS

Reviewer #1 (Remarks to the Author):

The paper by Gao and coll. reports the region of the CFTR protein that forms the binding site for CFTRinh-172, a broadly used compound that inhibits CFTR chloride channel activity. Authors used cryo-EM to identify the amino acid residues that specifically interact with the various parts of the CFTRinh-172 molecule. This information may pave the way for the development of more potent compounds, suitable as research tools and as drugs to treat human disorders involving CFTR hyperactivity. However, I think that some additional work should be done to support the conclusions of Authors. There are some results in the functional part of the study that do not fit with the structural data.

SPECIFIC COMMENTS

1) It is not clear which construct, wild type CFTR or E1371Q-CFTR, was used to perform functional experiments to define the effect of mutagenesis on CFTRinh-172 potency. I could not find this information. I think that experiments should be done on a small panel of selected mutations (e.g. K95A) introduced in either wild type and E1371Q background to check if and how the mutation of E1371 residue affects the effect of mutagenesis in other sites on inhibitor potency.

2) Authors cite a previous study (Stahl et al., 2012) which reported that CFTRinh-172 has a reduced inhibitory potency on CFTR from other species. That study compared human CFTR vs. CFTR from pig, killifish, and shark. Authors considered pig CFTR to understand the structural basis of the different potency of CFTRinh-172 on human and porcine CFTR. Experiments were done on the two variants of CFTR. There was indeed a different dose-response relationship (Figure 6) with a shift in IC₅₀ from 83 to 1850 nM (Table 1). Authors considered transmembrane domain 2 (TMD2) as a possible reason for the different behavior of the compound on the two CFTR proteins. Indeed, replacement of human TMD2 with pig TMD2 in human CFTR, shifted IC₅₀ even more (to 4023 nM). The key (unique) role of TMD2 in the interaction with CFTRinh-172 was further demonstrated by replacing pig TMD2 with human TMD2 in pig CFTR. In this case, the IC₅₀ became very low (55 nM). However, such results are puzzling since human and pig TMD2 are essentially identical. In particular, they both keep lysine 95 (K95) that was found by cryo-EM to interact with the carboxylic moiety of CFTRinh-172. Which region was precisely considered as TMD2 and used for the replacements? What is the reason for the shift in potency? By aligning human and pig CFTR, I could see only one amino acid difference, phenylalanine instead of isoleucine at position 87. Is this part of TMD2 and included in generation of the chimeric constructs? If so, Authors had to specifically mutagenize that residue to show that the results reproduced those obtained by replacing the entire TMD2. This part of the study is important since authors found that replacement of human TMD2 with pig TMD2 caused a shift in IC₅₀ even larger than that generated by K95 mutagenesis (from 83 nM to 852 nM for K95A).

Also Authors do not consider another study (Liu et al., *Am J Respir Cell Mol Biol* 36, 2007) in which CFTRinh-172 (and other CFTR inhibitors) was tested on human, pig, ferret, and mouse CFTR. Actually, ferret CFTR, not pig CFTR, was the most resistant to CFTRinh-172. Alignment of ferret and human TMD2 shows a more significant difference compared to pig TMD2, with an amino acid change at position 96. This study had to be considered in the evaluation of TMD2 structure-function relationship.

3) Authors do not cite other previous studies dealing with CFTRinh-172 mechanism of action. For example, Caci et al. (2008) which showed that mutagenesis of amino residues in a region forming the pore drastically lowered CFTRinh-172 potency. Also, Sonawane and Verkman (2008), which carried out a structure-activity relationship study, could have been considered to understand whether the moieties of CFTRinh-172 that were found important for inhibitory activity correspond to interactions with specific parts of CFTR protein.

4) As a control, a CFTR inhibitor having a completely different structure could have been used to check that potency was not affected by mutagenesis.

5) K95 is an amino acid residue that is part of the inner vestibule of CFTR pore and is directly involved in anion transport. Inhibition of CFTR by CFTRinh-172 is typically insensitive to transmembrane potential. How is this possible given that CFTRinh-172 is negatively charged and that, according to the present study, it interacts with a region of the protein that could be affected by the electric field?

Reviewer #2 (Remarks to the Author):

This work describes a structural study of the inhibition of CFTR by small molecule inhibitor Inh-172. Using cryo-electron microscopy, the authors determined the molecular structures of phosphorylated, ATP-bound E1371Q-CFTR successively in the absence and in the presence of the inhibitor to near-atomic resolution. By comparing the apo and holo structures, the authors identified the binding location of Inh-172 to be inside the pore near the bottleneck region. From molecular dynamics simulations of CFTR in the presence of the inhibitor in two possible orientations in the binding site identified by cryoEM, the authors picked the more stable binding pose 2 as the likely orientation of the drug. The CFTR residues interacting with the drug in this binding pose were used to guide subsequent experiments. Surprisingly, despite the similarity of the apo and holo structures, including in the inhibitor binding site, structural differences are noted in the extracellular mouth of the channel distal to the inhibitor binding site. These changes are proposed by the authors to be allosteric in nature.

Overall, the manuscript presents an important study of CFTR structure and activity modulation. The main finding of this study is the identification and the refinement of the inhibitor bound structure. However, the claim for an allosteric mechanism is not supported by the data shown; either the allosteric nature of inhibition by Inh-172 should be clarified or this claim should be dropped.

Major issues:

1. Although the authors refer to a recent study by Young et al. (BioRxiv, 2023) on the same topic, they do not discuss the contents of that study. It would be appropriate to compare and discuss the methodologies, the findings, and the conclusions of the two studies.

2. On page 10, the argument that the binding affinity of Inh-172 to CFTR depends on the conformational state of CFTR rests on the following assumptions: (i) that the drug would bind in exactly the same location and in the same pose in the closed state as it does in the open state, even though the geometry of the binding site is different; (ii) that the binding affinity would be decreased. This claim is not supported by the data and should be dropped, since supporting this claim would require extensive additional experimental and/or computational characterization (for example, by repeating the simulations after docking the inhibitor in the undimerized state).

As a related issue, on page 16, in support of the argument that Inh-172 binds with higher affinity in the NBD-dimerized state than in the NBD-separated state, the authors cited Kopeikin et al (J. Gen. Physiol., 2010), who found that recovery from inhibition occurs more slowly in the ATP-bound state. This may be due to the reduced accessibility of the binding site deep inside the pore in the NBD-dimerized state rather than to differences in the affinity of the binding site itself.

3. The authors should systematically quantify structural differences mentioned in the paper. In particular, although the overall RMSD between apo and holo structures is provided, this important information is missing in the comparison of the binding site structures, which are described as similar, and inversely, of helices in the extracellular mouth, where the structures diverge. The backbone and sidechain RMSD for specific regions of the protein should be provided wherever appropriate.

4. One would expect that the binding of ligands that act allosterically should involve structural differences in the binding site. As such, the physico-chemical basis of the allosteric mechanism

proposed by the authors is unclear, as the authors themselves admit on page 15. The authors should discuss what such alternatives to “the classical allosteric modulation mechanism” are and cite some precedents, if they exist, where no structural changes in the binding site were observed in an allosteric mechanism. MD simulations of the apo state successively without and with bound inhibitor and of the holo state following removal of the inhibitor could be performed to test the hypothesis that the conformational differences in the extracellular mouth of the pore are caused by inhibitor binding. Failing which, the claim for an allosteric mechanism remains speculative as it is not supported by the data provided.

Minor issues:

5. Please clarify the nature of the conformational changes observed in the extracellular portions of the TM helices. Do those changes include differences in overall helical tilts or changes in intrahelical kinks?

6. The experiment with human-pig chimeric CFTR is quite interesting as it suggests that TMD2 determines the sensitivity of CFTR to Inh-172. The authors should discuss how the three residues that were different between human and pig CFTR (Fig S4. Yellow highlighted) might alter the putative allosteric mechanism.

7. Please resolve the apparent contradiction between the following statements: “we found surprisingly negligible alterations between apo and holo structures” vs “we found Inh-172 induced major conformational changes in the extracellular mouth” (pages 10-11).

8. On page 15, the authors state that “our structural analysis may not reveal the supposed differences in binding interactions.” Please explain.

9. In Fig. 2b, the binding poses are hard to see and should be magnified for clarity.

10. On page 9, the authors stated that Inh-172 is coordinated by M348 and subsequently studied the mutant M348K based on this finding. Please specify the nature of that interaction.

11. On page 9, in the analysis of the salt-bridge interaction between K95 and Inh-172 carboxylate, please specify how the “charge-pair distance” was measured (e.g., as the distance between centers of charge of the ionic groups, between carboxylate oxygen and ammonium hydrogen atoms, or otherwise).

12. In Fig. 5, please specify the PDB codes next to the structures (instead of just phosphorylated/unphosphorylated etc) to avoid ambiguities, since there are currently many structures in either of these states.

Reviewer #3 (Remarks to the Author):

Gao, et al report the cryo-EM structure of CFTR bound to an inhibitor, Inh-172, along with MD simulations and electrophysiological assays to probe the interaction. They identify the binding pocket in the transmembrane region, near the pore, and propose an allosteric model for inhibitor mediated channel gating.

Overall, this manuscript feels a bit light on data and therefore a bit light on insights. The main advance seems to be the definition of the inh-172 binding site. Most of the work just clarifies and corroborates the slightly ambiguous ligand density from the cryoEM structure. However, the work that is presented appears to be well-executed and the major conclusions are generally well-supported by the data. In particular, the cryo-EM structure appears to be in good shape and I made just a few minor suggestions. The density for the ligand alone leaves some ambiguity regarding the bind pose, but is also supported by the MD and mutagenesis. Below are my detailed comments, which are generally minor.

MAJOR COMMENTS:

None

MINOR COMMENTS:

In 125-127; Fig 1a: The apparent MW of the complex is reported to be ~150 kDa by SEC, but no MW standards are shown in the figure.

Fig. 1: As these panels are primarily just demonstrating sample quality and that the authors have reproduced previous work, this could be moved to Extended Data

In 143-146: I feel like this comparison is a little inappropriate, as it seems to imply that, on the basis of similarity to a structure from a particular (well regarded!) lab, the new model is of high quality. The quality of the model should stand on its own. The previous sentence was adequate, so I suggest deleting this one.

In 160-161: Here the authors state that all 4 replicates are highly fluctuating, but visually only 2 stand out in Fig 2b; reps 3+4 for pose 1 look generally similar to the traces for pose 2 in Fig 2c. I would adjust the wording here to indicate that only trace 1+2 are highly fluctuating, or provide additional evidence (e.g., a quantitative metric of fluctuation) that indicates that 3+4 are also highly fluctuating.

In 193-207: The structures were not previously described as phosphorylated. It is a little confusing why this is suddenly mentioned here but not earlier. The subsequent comparisons were also a bit confusing, though I think I eventually got it. To me, some of the logic seems inverted, e.g., there was a previous observation that Inh-172 binds better to the open channel (In 209-210), therefore it makes sense to investigate whether the binding pocket changes between those states (instead of the current flow, which leads the reader to think there is an Inh-172-induced conformational change, which doesn't make sense and caused me to go in circles for 30 min...)

Fig. 6: This is an interesting initial finding, that swapping TMD2 between human and pig can more or less change the Inh-172 sensitivity. But I'm not sure it provides much insight into the mechanism, as it is such a large swap.

Throughout this manuscript, the dose response curves are not very well defined. e.g., in Fig 4, the WT curve has no points below ~35%. All the curves in this figure (and many other figures lack experiments at very low or high concentration, to firmly establish the upper and lower bounds. This could lead to a moderate left-right shift in the curves, depending on the experimental error. I'm not sure it is likely to change any major conclusion, but could certainly change the IC50s. In that regard, the degree of precision for the reported values in Table 1 seems to be a bit much (4 significant figures in some cases). For example, an IC50 of 4023 +/- 281 necessarily has uncertainty in the second digit. I could easily imagine these changing by a factor of 2 if repeated with more points, so again, what is the appropriate degree of precision?

Figs S5 and S6 appear to be only cited in Discussion, but introduce new data. They should be cited and discussed in the results, or removed.

Table S1: It is nice to include PDB, EMDB, and EMPIAR codes in the table for easy reference. I would also suggest including map-model CC values (mask, box, peaks, and volume).

PDB reports indicate 5 residues that don't match the reference sequence of CTFR. What is the source of these additional mutations?

We thank all the reviewers for their comments. Below are the point-by-point responses to all the comments. The comments are in black, our responses are in blue and all corresponding changes in the manuscript text are in red.

Reviewer #1 (Remarks to the Author):

The paper by Gao and coll. reports the region of the CFTR protein that forms the binding site for CFTRinh-172, a broadly used compound that inhibits CFTR chloride channel activity. Authors used cryo-EM to identify the amino acid residues that specifically interact with the various parts of the CFTRinh-172 molecule. This information may pave the way for the development of more potent compounds, suitable as research tools and as drugs to treat human disorders involving CFTR hyperactivity. However, I think that some additional work should be done to support the conclusions of Authors. There are some results in the functional part of the study that do not fit with the structural data.

SPECIFIC COMMENTS

1) It is not clear which construct, wild type CFTR or E1371Q-CFTR, was used to perform functional experiments to define the effect of mutagenesis on CFTRinh-172 potency. I could not find this information. I think that experiments should be done on a small panel of selected mutations (e.g. K95A) introduced in either wild type and E1371Q background to check if and how the mutation of E1371 residue affects the effect of mutagenesis in other sites on inhibitor potency.

We apologize for not making this clear in the original manuscript. All our functional experiments were conducted in the wild-type background, and we now have included a sentence on page 9 to make this clear for the readers.

A reliable dose-response relationship in patch-clamp experiments depends critically on our ability to determine an accurate baseline current. For wild-type CFTR, this can be accomplished by simply removing ATP from the perfusate—the current will decay within seconds upon ATP depletion. However, the non-hydrolytic closure of E1371Q-CFTR upon ATP washout is extremely slow, with a time constant of more than 100 seconds. As the quality of the membrane patch usually deteriorates over time, this prolonged waiting time is prohibitable for a more reliable baseline measurement. In theory, this difficulty can be overcome by using a high concentration of Inh-172, but the E1371Q mutation by itself poses an even more serious challenge for an accurate determination of the dose-response relationship. As reported in Kopeikin et al., (2010), maneuvers that stabilize the open state by abolishing ATP hydrolysis (e.g., E1371Q) drastically increased the apparent affinity of Inh-172. That means picomolar concentrations of Inh-172 would be needed at the lower bound of the dose-response curve for E1371Q-CFTR. At such low concentrations, the slow diffusion rate of the drug lengthens the time needed for the current to achieve a steady state, and the experimental timeframe would again be drastically prolonged (please also see the response to Reviewer 3's comment #7 related to this technical challenge). Therefore, we deem the E1371Q background not ideal for

our functional studies. Of note, a similar approach has been adopted for Inh-172 by Young et al. (Young et al., 2023) as well as for CFTR potentiators (Liu et al., 2019) where cryo-EM structures were determined with E1371Q-CFTR but all functional tests were carried out in the WT background.

2) Authors cite a previous study (Stahl et al., 2012) which reported that CFTRinh-172 has a reduced inhibitory potency on CFTR from other species. That study compared human CFTR vs. CFTR from pig, killifish, and shark. Authors considered pig CFTR to understand the structural basis of the different potency of CFTRinh-172 on human and porcine CFTR. Experiments were done on the two variants of CFTR. There was indeed a different dose-response relationship (Figure 6) with a shift in IC₅₀ from 83 to 1850 nM (Table 1). Authors considered transmembrane domain 2 (TMD2) as a possible reason for the different behavior of the compound on the two CFTR proteins. Indeed, replacement of human TMD2 with pig TMD2 in human CFTR, shifted IC₅₀ even more (to 4023 nM). The key (unique) role of TMD2 in the interaction with CFTRinh-172 was further demonstrated by replacing pig TMD2 with human TMD2 in pig CFTR. In this case, the IC₅₀ became very low (55 nM). However, such results are puzzling since human and pig TMD2 are essentially identical. In particular, they both keep lysine 95 (K95) that was found by cryo-EM to interact with the carboxylic moiety of CFTRinh-172. Which region was precisely considered as TMD2 and used for the replacements? What is the reason for the shift in potency? By aligning human and pig CFTR, I could see only one amino acid difference, phenylalanine instead of isoleucine at position 87. Is this part of TMD2 and included in generation of the chimeric constructs? If so, Authors had to specifically mutagenize that residue to show that the results reproduced those obtained by replacing the entire TMD2. This part of the study is important since authors found that replacement of human TMD2 with pig TMD2 caused a shift in IC₅₀ even larger than that generated by K95 mutagenesis (from 83 nM to 852 nM for K95A).

Also Authors do not consider another study (Liu et al., Am J Respir Cell Mol Biol 36, 2007) in which CFTRinh-172 (and other CFTR inhibitors) was tested on human, pig, ferret, and mouse CFTR. Actually, ferret CFTR, not pig CFTR, was the most resistant to CFTRinh-172. Alignment of ferret and human TMD2 shows a more significant difference compared to pig TMD2, with an amino acid change at position 96. This study had to be considered in the evaluation of TMD2 structure-function relationship.

Since the identification of the *cftr* gene (Riordan et al., 1989), the domain structure of CFTR was traditionally defined as follows (from the N- to the C- terminus): TMD1, NBD1, R domain, TMD2, and NBD2. TMD2 consists of six transmembrane segments (TM)—from TM7 to TM12—with amino acid residues from 844 to 1172 in human CFTR. In **Figure S4**, the alignment of human TMD2 and pig TMD2 showed numerous amino acid differences indicated by period (.) and colon (:). The reviewer's remark that "such results are puzzling since human and pig TMD2 are essentially identical" might be due to an erroneous definition of TMD2.

However, what was puzzling to us is that while the binding site residues for Inh-172 (**Figure S4**, green) are well conserved in pig CFTR, pig CFTR shows much lower sensitivity to Inh-172 compared with human CFTR (IC₅₀ of 83 nM for human CFTR vs. 1850 nM for pig CFTR). We,

therefore, conducted domain swap and finer transmembrane segment swap experiments to understand what accounts for its lower sensitivity.

The reviewer noticed an amino acid difference at position 87 between human (phenylalanine) and pig CFTR (isoleucine). This residue is in TMD1 not in TMD2. In this revision, we further show that swapping TMD1 between human and pig CFTR does not affect the orthologs' sensitivities to Inh-172 (see updated **Figure 6**). Moreover, position 87 is not in the binding site either. It is unlikely that the difference at position 87 will account for pig CFTR's reduced sensitivity to Inh-172.

As for ferret CFTR, we have now cited the study showing its different sensitivity to Inh-172 in the revised manuscript on page 11. Sequence alignment of human CFTR and ferret CFTR shows that the binding site residue S1141 (human CFTR) is instead a glycine in ferret CFTR (G1143, highlighted in yellow, see figure below). All the other binding site residues are well conserved (highlighted in green). Whether this amino acid difference, or the alanine-to-serine change at position 96 brought up by the reviewer, or any other differences between human and ferret orthologs could account for ferret CFTR's lowered affinity for Inh-172 warrant rigorous examinations, which are way beyond the scope of the current study.

Human CFTR	1	MQRSPLEKASVSKLFFSWTRPILRKGYRQRLELSDIYQIPSVDSADNLS	Human CFTR	750	FRISVISTGFTLQARRRQSVLNMTHSV-NQGQNIHRKTTASTRKVSLAP
Ferret CFTR	1	MQRSPLEKASVSKLFFSWTRPILRKGYRQRLELSDIYQIPSDSADNLS	Ferret CFTR	751	FRSNMINTGFTLQRRQRQSVLNMTCSPGNGQGSFHGRASSTRMSLAP
Human CFTR	51	EKLEREWRELASKKNKLNALRRCFWRFMFYGIPLYLGEVTVQPL	Human CFTR	799	QANLTELDIYSRRLSQETGLEISEEINEEDLKECLFDDMESIPAVTTWNT
Ferret CFTR	51	EKLEREWRELASKKNKLNALRRCFWRFMFYGIPLYLGEVTVQPL	Ferret CFTR	801	QANLTEMDIYSRRLSQDSGLEISEEINEEDLKECFIDVDESIFPVTWNT
Human CFTR	101	LLGRIIASYDPDKKEERSIAYLIGLGLCLLIVRTLLHPAIFGLHHIGM	Human CFTR	849	YLRVITVHKSLFLVLIWCLVIFLAEVAASLVVLLWNTPLQKQGNSTHS
Ferret CFTR	101	LLGRIIASYDPDKKEERSIAYLIGLGLCLLIVRTLLHPAIFGLHHIGM	Ferret CFTR	851	YLRVITVHKSLFLVLIWCLVIFLAEVAASLVVLLWNTPLQKQGNSTQS
Human CFTR	151	QMRIAMFSLIYKTKLKSRLVDKISIGQLVSLNKNKFDDEGLALAHF	Human CFTR	899	RNNSYAVIITSTSSYVYFYIYVTLAMGFFRGLPLVHTLIIVSKIL
Ferret CFTR	151	QMRIAMFSLIYKTKLKSRLVDKISIGQLVSLNKNKFDDEGLALAHF	Ferret CFTR	901	INSSYTVIPTSSTSYVYFYIYVTLALGFFRGLPLVHTLIIVSKIL
Human CFTR	201	VWIAPLQVALLMGLIWLQASAFQGLFLVLAQAGLGRMMKMYRQD	Human CFTR	949	HHKMLHSLVLAQPMSTLNLKAGGILNRFSDIAIIDLDDLLPLTIFDFIQLL
Ferret CFTR	201	VWIAPLQVLLMGLLWLLQASAFQGLFLVLAQAGLGRMMKMYRQD	Ferret CFTR	951	HHKMLHSLVLAQPMSTLNLKAGGILNRFSDIAIIDLDDLLPLTIFDFVQLL
Human CFTR	251	RAGKISERLIVITSEMIENIQSVKAYWEAEAMEKMIENLRQTELKTRKAA	Human CFTR	999	LVIQIAVAVLQPIYFVATVPIVAFIMLRAYFLQTSQQQLKQLESEGR
Ferret CFTR	251	RAGKINERLIVITSEMIENIQSVKAYWEAEAMEKMIENLRQTELKTRKAA	Ferret CFTR	1001	LVIQIAVAVLQPIYFVATVPIVAFIMLRAYFLHTSQQLKQLESEGR
Human CFTR	301	VYRYFNSSA FSGFFVFLSVLPYALIKGILLRKFITFTISFCVLRFAV	Human CFTR	1049	SPIFTHLVTSLKGLWTLRAFGRQPYFETLPHKALNLTANWFLYLSLRAW
Ferret CFTR	301	VYRYFNSSA FSGFFVFLSVLPYALIKTIVLRKFITFTISFCVLRFAV	Ferret CFTR	1051	SPIFTHLVTSLKGLWTLRAFGRQPYFETLPHKALNLTANWFLYLSLRAW
Human CFTR	351	TROFFWAVQTYWDSLGAINKIQDFLQKQEKYKLEYNLTTEVVMENVTAF	Human CFTR	1099	FQMRIEMIFVIFIAVTFISILITTEGEGRQVGIILTLAMSTLQAVN
Ferret CFTR	351	TROFFWAVQTYWDSLGAINKIQDFLQKQEKYKLEYNLTTEVVMENVTAF	Ferret CFTR	1101	FQMRMEIIFVIFIAVTFISILITTEGEGVAVGVIILTLAMSTLQAVN
Human CFTR	401	WEEGFGELFEKAKQNNNRKTSNGDDSLFNSFLGTPVLKIDINFKIER	Human CFTR	1149	SSIDVDSLMSRSRVFKFIDMPTEGK--PTKSTKPYKNGQLSKVMIENS
Ferret CFTR	401	WEEGFGELLEKAKQNSNDRKISNADNSLFFNSFLGAPVLKDISFKIER	Ferret CFTR	1151	SSIDVDSLMSRSRVFKFIDMPAEESKPTKSFKPSKDVQLSKVITENH
Human CFTR	451	QQLLAVAGSTGAGKTSLLMMIMGELEPSEGKIKHSGRISFCSQPSWIMPG	Human CFTR	1197	HVKKDDIWPSSGGQMTVKDLTKARYTEGGNAILENISFSISPGQVGLLGR
Ferret CFTR	451	QQLLAVAGSTGAGKTSLLMMIMGELEPSEGKIKHSGRISFCSQPSWIMPG	Ferret CFTR	1201	HVREDDIWPSSGGQMTVKDLTKARYIDGGNAILENISFSISPGQVGLLGR
Human CFTR	501	TIKENIIFGVSYDEYRYSVIAKQCLEEDIISKFAEKDNIVLGEGGITLSG	Human CFTR	1247	GSGKSTLLSAFLRLNTEGEIQIDGVSWDSITLQQRKAFQVIGPKVFI
Ferret CFTR	501	TIKENIIFGVSYDEYRYSVIAKQCLEEDIISKFAEKDNIVLGEGGITLSG	Ferret CFTR	1251	GSGKSTLLSAFLRLNTEGEIQIDGVSWDSITLQQRKAFQVIGPKVFI
Human CFTR	551	QQRARISLARAVYKDADLVLLDSPFGYLDVLTKEIIFESCVCCKLMANKTR	Human CFTR	1297	SGTFRKNDLDPYEQWSDQEIWKVADEVGLRSVIEQFPGLKDFVLVDGGCVL
Ferret CFTR	551	QQRARISLARAVYKDADLVLLDSPFGYLDVLTKEIIFESCVCCKLMANKTR	Ferret CFTR	1301	SGTFRKNDLDPYQWSDQEIWKVADEVGLRSVIEQFPGLKDFVLVDGGCVL
Human CFTR	601	ILVTSKMEHLKAKDKILILNEGSSYFYGTFSLQNLQDFDSSKLMGCDSE	Human CFTR	1347	SHGHKQMLCLARSVLSKAKILLLDEPSAHLDPITYQIIRRTLKQAFADCT
Ferret CFTR	601	ILVTSKMEHLKAKDKILILNEGSSYFYGTFSLQNLQDFDSSKLMGYDSE	Ferret CFTR	1351	SHGHKQMLCLARSVLSKAKILLLDEPSAHLDPITYQIIRRTLKQAFADCT
Human CFTR	651	DQFSAERRNSIITETLRRFSLEGDAFVSNWTEKTKQSFKQVGEFGEKRNNS	Human CFTR	1397	VILCEHRIEAMLECCQFLVIEENKVRQYDSIQKLNERSLFRQAISPSDR
Ferret CFTR	651	DQFSAERRNSIITETLRRFSLEGDAFVSNWTEKTKQSFKQVGEFGEKRNNS	Ferret CFTR	1401	VILSEHRIEAMLECCQFLVIEENKVRQYDSIQRLLSSEKLSFRQAISPSDR
Human CFTR	701	ILNPINSIRKFSIVQKTPQLQMGIEEDSDEFLERRLSLVPD-SEQGEAIL	Human CFTR	1447	VKLFPHRNSKCKSKPQIAALKEETEVEEQDTRL 1480
Ferret CFTR	701	ILNPNVNSIRKFSVQKTPQLQMGIEEDSDEFLERRLSLVPDGAEQGEAIL	Ferret CFTR	1451	LRFFPHRNSKHKRSRQIAALKEETEVEEQETRL 1484

3) Authors do not cite other previous studies dealing with CFTRinh-172 mechanism of action.

For example, Caci et al. (2008) which showed that mutagenesis of amino residues in a region forming the pore drastically lowered CFTRinh-172 potency. Also, Sonawane and Verkman (2008), which carried out a structure-activity relationship study, could have been considered to understand whether the moieties of CFTRinh-172 that were found important for inhibitory activity correspond to interactions with specific parts of CFTR protein.

We thank the reviewer for pointing this out. Now, these two papers are both cited, and their contents are incorporated into the 2nd and 10th paragraph of the Discussion.

4) As a control, a CFTR inhibitor having a completely different structure could have been used to check that potency was not affected by mutagenesis.

Most CFTR inhibitors block CFTR pore from the cytoplasmic side in a voltage-dependent manner, such as glibenclamide (Sheppard and Robinson, 1997), NPPB (Walsh et al., 1999), and DNDS (Linsdell and Hanrahan, 1996). Since these CFTR inhibitors likely block the open channel by traveling down the internal vestibule to the narrow region of the pore, the mutations introduced to the Inh-172's binding site, which is just below the narrow region, would inevitably affect the actions of these CFTR blockers. Indeed, neutralizing K95 has been shown to affect the voltage-dependent block of glibenclamide, NPPB, DNDS, and other CFTR blockers (reviewed in Linsdell, 2014). Hence, these large anionic CFTR inhibitors are poor candidates as a negative control.

On the other hand, smaller CFTR inhibitors that are structurally different from Inh-172 are more suitable for the experiment suggested by the reviewer. We used thiocyanate (SCN⁻) as the negative control to check if its potency is affected by mutagenesis. To simplify data interpretation, we compared the potency of SCN⁻ between E1371Q and E1371Q/W1145Y-CFTR. The E1371Q background eliminates any potential gating effects of SCN⁻, and the conserved W-to-Y mutation minimized the structural change. As seen in the figure below, the potency of SCN⁻ (percent of the blockade) is barely affected by the mutation.

Of note, we unexpectedly found that the I-V relationship of W1145Y/E1371Q-CFTR is outwardly rectified, suggesting that the W-to-Y mutation alters CFTR's chloride conduction. This observation is a cautionary tale that the effects of mutagenesis near CFTR's pore region could be complicated, and that identification of a drug-binding site should be rooted in converging evidence from both structural and functional data, which is why we strived to integrate cryo-EM, MD simulation and electrophysiology techniques to affirm the binding site for Inh-172.

Comparison of intracellular SCN⁻ blockade on E1371Q- and W1145Y/E1371Q-CFTR. (A and B) Representative I-V relationships of E1371Q- (A) and W1145Y/E1371Q-CFTR (B) in response to cytoplasmic 10 mM SCN⁻. The I-V relationships were obtained by averaging five \pm 100 mV voltage ramps on the same excised patch. (C) Fraction of block by 2 mM or 10 mM SCN⁻ on E1371Q-CFTR (black, n = 3) and W1145Y/E1371Q-CFTR (red, n = 3). Fraction of block was calculated according to the equation “Fraction of block = $1 - (I_{SCN^-})/I_0$ ” where I_{SCN^-} and I_0 represent the blocked and unblocked current at a given voltage on the same patch. (D) Averaged fraction of block by 2 mM or 10 mM SCN⁻ from (C). Note that the potency of SCN⁻ does not differ between E1371Q- and W1145Y/E1371Q-CFTR.

5) K95 is an amino acid residue that is part of the inner vestibule of CFTR pore and is directly involved in anion transport. Inhibition of CFTR by CFTRinh-172 is typically insensitive to transmembrane potential. How is this possible given that CFTRinh-172 is negatively charged and that, according to the present study, it interacts with a region of the protein that could be affected by the electric field?

Indeed, as the reviewer pointed out, the voltage-independent inhibition by Inh-172 of CFTR was observed by both our group (Kopeikin et al., 2010) and Verkman group (Ma et al., 2002). To exhibit voltage-dependent blocking, a channel blocker needs to experience a voltage drop across the electric field across the cell membrane (classical Woodhull model). We reason that for CFTR, the voltage drop should happen at the narrow region close to the extracellular end of its pore. Inh-172 binds not at this narrow region but at a site more intracellular to the narrow

region, facing CFTR's wide inner vestibule (see the figure below). As the inner vestibule is spatial enough to accommodate ~180 water molecules (Hwang et al., 2018), this region should be considered a low-resistance part of the pore—hence little voltage drop. That is, Inh-172 would not experience a steep voltage drop on its way to its binding site; hence its inhibition is voltage-independent. We are cautious not to include this in our discussion since the mechanism of CFTR inhibition by Inh-172 involves not simply binding to the identified binding site; whether the conformational changes following Inh-172 binding are affected by membrane potential is unknown at this juncture.

Reviewer #2 (Remarks to the Author):

This work describes a structural study of the inhibition of CFTR by small molecule inhibitor Inh-172. Using cryo-electron microscopy, the authors determined the molecular structures of phosphorylated, ATP-bound E1371Q-CFTR successively in the absence and in the presence of the inhibitor to near-atomic resolution. By comparing the apo and holo structures, the authors identified the binding location of Inh-172 to be inside the pore near the bottleneck region. From molecular dynamics simulations of CFTR in the presence of the inhibitor in two possible orientations in the binding site identified by cryoEM, the authors picked the more stable binding pose 2 as the likely orientation of the drug. The CFTR residues interacting with the drug in this binding pose were used to guide subsequent experiments. Surprisingly, despite the similarity of the apo and holo structures, including in the inhibitor binding site, structural differences are noted in the extracellular mouth of the channel distal to the inhibitor binding site. These changes are proposed by the authors to be allosteric in nature.

Overall, the manuscript presents an important study of CFTR structure and activity modulation. The main finding of this study is the identification and the refinement of the inhibitor bound structure. However, the claim for an allosteric mechanism is not supported by the data shown; either the allosteric nature of inhibition by Inh-172 should be clarified or this claim should be dropped.

Major issues:

1) Although the authors refer to a recent study by Young et al. (BioRxiv, 2023) on the same topic, they do not discuss the contents of that study. It would be appropriate to compare and discuss the methodologies, the findings, and the conclusions of the two studies.

We added these comparisons in the first paragraph of the Discussion.

2) On page 10, the argument that the binding affinity of Inh-172 to CFTR depends on the conformational state of CFTR rests on the following assumptions: (i) that the drug would bind in exactly the same location and in the same pose in the closed state as it does in the open state, even though the geometry of the binding site is different; (ii) that the binding affinity would be decreased. This claim is not supported by the data and should be dropped, since supporting this claim would require extensive additional experimental and/or computational characterization (for example, by repeating the simulations after docking the inhibitor in the undimerized state).

We agree with the reviewer that there is no direct evidence to show that in the undimerized state Inh-172 will interact with the same residues, bind in exactly the same location, and adopt the same pose as it does in the NBDs dimerized state. Therefore, we moved **Figure 5c and d** (now **Figure S5**) to the Discussion as a possible explanation for our observation that the open state of CFTR has a higher affinity for Inh-172 than the closed state.

As a related issue, on page 16, in support of the argument that Inh-172 binds with higher affinity in the NBD-dimerized state than in the NBD-separated state, the authors cited Kopeikin et al. (J. Gen. Physiol., 2010), who found that recovery from inhibition occurs more slowly in the ATP-bound state. This may be due to the reduced accessibility of the binding site deep inside the pore in the NBD-dimerized state rather than to differences in the affinity of the binding site itself.

We agree with the reviewer that NBD dimerization may slow the recovery because the binding site is now deep 'wrapped' by two TMDs. If dimerized NBDs indeed prevent access of Inh-172 to its binding site, both the on and off rates should be affected with little change of the affinity. However, our previous work showed that the dose-response curve shifts to the left drastically when the channel is locked open (Kopeikin et al., 2010). Of note, we have previously shown that a much larger molecule glibenclamide (molecular weight 494.0 g/mol vs 409.4 g/mol of Inh-172) has been shown to readily get in and out of the pore of "lock-opened" CFTR with dimerized NBDs at an off-rate of $\sim 10 \text{ s}^{-1}$ (Figs. 5 and 6 in Zhou et al., 2002). Therefore, this mechanism is less likely to explain a slow recovery (in the range of tens to hundreds of seconds) from Inh-172-induced inhibition. The extremely slow recovery rate upon washout of Inh-172 more likely reflects the intrinsic stability of Inh-172-bound "inactivated" state.

3) The authors should systematically quantify structural differences mentioned in the paper. In particular, although the overall RMSD between apo and holo structures is provided, this

important information is missing in the comparison of the binding site structures, which are described as similar, and inversely, of helices in the extracellular mouth, where the structures diverge. The backbone and sidechain RMSD for specific regions of the protein should be provided wherever appropriate.

Per the reviewer's request, we have now included the RMSD of the binding site residues (as depicted in **Figure 3a**) between the Inh-172-free structure and the Inh-172-bound structure. The RMSD of the extracellular part of TMs 1, 8, and 12 between the two structures is also indicated in the text.

4) One would expect that the binding of ligands that act allosterically should involve structural differences in the binding site. As such, the physico-chemical basis of the allosteric mechanism proposed by the authors is unclear, as the authors themselves admit on page 15. The authors should discuss what such alternatives to “the classical allosteric modulation mechanism” are and cite some precedents, if they exist, where no structural changes in the binding site were observed in an allosteric mechanism. MD simulations of the apo state successively without and with bound inhibitor and of the holo state following removal of the inhibitor could be performed to test the hypothesis that the conformational differences in the extracellular mouth of the pore are caused by inhibitor binding. Failing which, the claim for an allosteric mechanism remains speculative as it is not supported by the data provided.

We agree with the reviewer that a classic allosteric modulation reaction involves conformational change(s) at the binding site of the modulators. We were thus surprised to see that those amino acids are relatively static after binding of Inh-172. However, conformational differences at the extracellular regions of TMs 1, 8, and 12, which are distal from where Inh-172 binds, are indeed observed. In this sense, the binding of Inh-172 causes a conformational change in other parts of the protein, and this change subsequently obstructs the ion permeation pathway—hence an allosteric regulation. Having said that, we still could not exclude the possibility that the cryo-EM structures may not reveal the fine conformational differences at the binding site because of resolution limitation or information loss when converting the electron density map to molecular structures where manual intervention is inevitable.

However, this puzzling observation—lack of structural change in the binding site upon drug binding—is not unprecedented in the CFTR field. For example, when CFTR/VX-770 and CFTR/GLPG-1837 structures were determined, little structural difference was observed between the drug-free and drug-bound structures (Liu et al., 2019). However, it is well established that these two potentiators modulate CFTR activity by an allosteric modulation mechanism (Yeh et al., 2017).

Taking into consideration the reviewer's concern about the proper definition of a classic allosteric modulation model, we now include a more thorough discussion to further clarify that we do not see obvious structural changes in the binding site upon Inh-172 binding, which

deviates from the classic allosteric modulation mechanism (See the fifth paragraph of Discussion on page 16).

Minor issues:

5) Please clarify the nature of the conformational changes observed in the extracellular portions of the TM helices. Do those changes include differences in overall helical tilts or changes in intrahelical kinks?

These major conformational changes include 1) the extracellular end of TM1 (from V93 to Y109) tilts outward from the central axis for about 10°; 2) TM8 (from T910 to L926) tilts outward from the central axis for about 13°; 3) TM12 (from V1129 to L1143) wedges into the central axis with a 30° tilt. TM8 is the only TM that has a kink from L927 to G934 and we did not see much structural change as the full atom RMSD for these residues between the two structures is 1.4 Å. Now we have included the tilting angles for each TM in the Results of the manuscript (see updated **Figure 5**).

6) The experiment with human-pig chimeric CFTR is quite interesting as it suggests that TMD2 determines the sensitivity of CFTR to Inh-172. The authors should discuss how the three residues that were different between human and pig CFTR (Fig S4. Yellow highlighted) might alter the putative allosteric mechanism.

For the three different residues in TM8, while position 925 is located at the extracellular half of TM8, positions 929 and 931 are part of the kink in TM8, which has been proposed to serve as a pivot for the motion of the extracellular half of TM8 during CFTR gating (Zhang et al., 2017). In our hand, the motion of TM8 induced by Inh-172 takes place right above the kink, therefore, the kink may also serve as a hinge for Inh-172-induced conformational changes in TM8. By aligning the CFTR sequences of different orthologs, we realized that the T925 is well-conserved for many CFTR species, but not for pig CFTR (glycine 925, see the figure below). This threonine-to-glycine change alone may not play a major role in lowering the sensitivity of pig CFTR towards Inh-172 as the ferret CFTR, which is even less sensitive to Inh-172 than pig CFTR (Liu et al., 2007), like human CFTR also has a threonine at this position (see above figure in response to the second comment from reviewer 1 (Liu et al., 2007)).

Regarding the methionine to leucine change at position 929 and phenylalanine to leucine change at position 931, it is possible that different side chains introduce different dynamics in their local environment and hence influence the motion of TM8 as they are in the critical kink region, especially given that methionine and phenylalanine are more likely to participate in specific sidechain chemistry (e.g., methylthio- and π -orbital interactions, respectively) than leucine (which is basically hydrophobic). Noticeably, the M to L change at position 929 and F to L change at position 931 are observed among many different CFTR orthologs (see the figure below). While it is difficult to specify the exact reason for each species, our data of the TMD2 swap experiments suggest TMD2 as a whole should account for the different sensitivities of CFTR orthologs to Inh-172 as the allosteric modulation is most likely done beyond a few specific

individual residues level (see our #6 response to reviewer #3 below). Now we added a few sentences on this topic into the 9th paragraph in Discussion.

	961	1000
HumanCFTR (903)	YAVIIITSTSSYYVFYIYVGVADTLLA	MGFFRGLPLVHTLI
chimp CFTR-WWR (903)	YAVIIITSTSSYYVFYIYVGVADTLLA	MGFFRGLPLVHTLI
Orangutan CFTR (903)	YAVIIITSTSSYYVFYIYVGVADTLLA	MGFFRGLPLVHTLI
gorilla CFTR (903)	YAVIIITSTSSYYVFYIYVGVADTLLA	MGFFRGLPLVHTLI
Japanese macaque (903)	YAVIIITRTSSYYVFYIYVGVADTLLA	MGFFRGLPLVHTLI
olive baboon (903)	YAVIIITRTSSYYVFYIYVGVADTLLA	MGFFRGLPLVHTLI
crab-eating macaque (903)	YAVIIITRTSSYYVFYIYVGVADTLLA	MGFFRGLPLVHTLI
rhesus (903)	YAVIIITRTSSYYVFYIYVGVADTLLA	MGFFRGLPLVHTLI
pig-tailed macaque (903)	YAVIIITRTSSYYVFYIYVGVADTLLA	MGFFRGLPLVHTLI
vervet CFTR (903)	YAVIIITRTSSYYVFYIYVGVADTLLA	MGFFRGLPLVHTLI
MarmosetCFTR (903)	YAVIIITNTSSYYVFYIYVGVADTLLA	LGFRRGLPLVHTLI
Mouse lemur CFTR (903)	YAVIIITSTSSYYVFYIYVGVADTLLA	LGFRRGLPLVHTLI
lemurcftr-WWR (903)	YAVIIITSTSSYYVLYIYVGVADTLLA	LGFRRGLPLVHTLI
Galago CFTR (904)	SAVIIITSTSSYYLFYIYVGVADTLLA	LGFRRGLPLVHTLI
BovineCFTR (903)	YAVIIITSTSSYYIFYIYVGVADTLLA	LGLFRGLPLVHTLI
SheepCFTR (903)	YAVIIITSTSSYYIFYIYVGVADTLLA	LGLFRGLPLVHTLI
PigCFTR WWR (904)	YAVIIITSTSSYYVFYIYVGVADGLLA	LGLFRGLPLVHTLI
horseCFTR-wwr (903)	YAVIIITSTSSYYVFYIYVGVADTLLA	LGLFRGLPLVHTLI
DogCFTR WWR (904)	YAVIITSTSSYYIFYIYVGVADTLLA	LGLFRGLPLVHTLI
CatCFTR WWR (902)	YAVIITNTSTYYVFYIYVGVADTLLA	LGFRRGLPLVHTLI
rabbitCFTR (903)	YAVIIITNTSSYYVFYIYVGVADTLLA	LGLFRGLPLVHTLI
African hedgehogCFTR (905)	YGVIIITNTSSYYIYIYVGVADTLLA	LGLRRGLPLVHTLI

7) Please resolve the apparent contradiction between the following statements: “we found surprisingly negligible alterations between apo and holo structures” vs “we found Inh-172 induced major conformational changes in the extracellular mouth” (pages 10-11).

Sorry for causing such confusion. With ‘we found surprisingly negligible alterations between apo and holo structures,’ we were referring to the binding site residues as indicated by the adverbial modifier immediately before these words. With ‘we found Inh-172 induced major conformational changes in the extracellular mouth,’ we were referring to the extracellular segments of TMs 1, 8, and 12. We have now modified the text to avoid confusion.

8) On page 15, the authors state that “our structural analysis may not reveal the supposed differences in binding interactions.” Please explain.

As in response to the reviewer’s major comment #4, the resolution limitation could be a potential reason why no conformational differences were detected. With a higher resolution, say at 1 Å, the assignment of each side chain of those binding site residues could be different from what we observe now. This technical limitation arises from the way how cryo-EM maps are calculated. The algorithm averages tens of thousands of particles, all of which may or may not be in the same conformation, into a final map. It is possible that in some particles the side chain information is different from what we see with the final structure. Another potential reason that the structure is not able to 100% reflect the information carried in the cryo-EM map, or further, the dynamic taking place in the protein, is that there is inevitable loss/alteration when the information passes from the electron density maps to molecular structures which are built with manual adjustments based on the maps. All these reasons could

account for imprecision in information collection and interpretation. However, the structures presented in our work have gone through multiple rounds of refinement and adjustment to make sure we have, to our best, presented our data judiciously. We have now elaborated on this point.

9) In Fig. 2b, the binding poses are hard to see and should be magnified for clarity.

We enlarged the binding pose for both Fig. 2b and Fig. 2c in the updated **Figure 2**.

10) On page 9, the authors stated that Inh-172 is coordinated by M348 and subsequently studied the mutant M348K based on this finding. Please specify the nature of that interaction.

We have added the 'hydrophobic interaction' in the text as the reviewer requested.

11) On page 9, in the analysis of the salt-bridge interaction between K95 and Inh-172 carboxylate, please specify how the "charge-pair distance" was measured (e.g., as the distance between centers of charge of the ionic groups, between carboxylate oxygen and ammonium hydrogen atoms, or otherwise).

The salt-bridge interaction was measured based on the distance between the K95 NZ (nitrogen zeta) atom and the center of mass of Inh-172 carboxylate oxygen atoms. This has been updated in the manuscript.

12) In Fig. 5, please specify the PDB codes next to the structures (instead of just phosphorylated/unphosphorylated etc) to avoid ambiguities, since there are currently many structures in either of these states.

We have added the PDB codes in the new **Figure S5**.

Reviewer #3 (Remarks to the Author):

Gao, et al report the cryo-EM structure of CFTR bound to an inhibitor, Inh-172, along with MD simulations and electrophysiological assays to probe the interaction. They identify the binding pocket in the transmembrane region, near the pore, and propose an allosteric model for inhibitor mediated channel gating.

Overall, this manuscript feels a bit light on data and therefore a bit light on insights. The main advance seems to be the definition of the inh-172 binding site. Most of the work just clarifies and corroborates the slightly ambiguous ligand density from the cryoEM structure. However, the work that is presented appears to be well-executed and the major conclusions are generally well-supported by the data. In particular, the cryo-EM structure appears to be in good shape and I made just a few minor suggestions. The density for the ligand alone leaves some ambiguity regarding the bind pose, but is also supported by the MD and mutagenesis. Below are my detailed comments, which are generally minor.

MAJOR COMMENTS:

None

MINOR COMMENTS:

1) In 125-127; Fig 1a: The apparent MW of the complex is reported to be ~150 kDa by SEC, but no MW standards are shown in the figure.

We thank the reviewer for pointing out our mistake. The elution at ~15 ml position corresponds to the CFTR protein which has a molecular weight of ~150 kDa, but it is the SDS-PAGE gel that shows the molecular weight of the eluted protein. We have revised the statement in the text now to accurately represent what is shown in **Figure 1**.

2) Fig. 1: As these panels are primarily just demonstrating sample quality and that the authors have reproduced previous work, this could be moved to Extended Data

We agree with the reviewer that this is usually the case for a study that utilizes a well-established protein source. While the BacMam vector was used for all previous structural studies of CFTR by Prof. Jue Chen's lab, here a different protein production system using a lentiviral vector apparently can produce CFTR proteins with equally good quality for both functional and structural study. As our manuscript will be the first to document the high-resolution cryo-EM structures determined for CFTR proteins purified with this lentiviral system, we feel it appropriate to keep this figure in the main text to raise public interest in this alternative system.

3) In 143-146: I feel like this comparison is a little inappropriate, as it seems to imply that, on the basis of similarity to a structure from a particular (well regarded!) lab, the new model is of high quality. The quality of the model should stand on its own. The previous sentence was adequate, so I suggest deleting this one.

The reviewer's point is well-taken. We have removed the words indicated by the reviewer.

4) In 160-161: Here the authors state that all 4 replicates are highly fluctuating, but visually only 2 stand out in Fig 2b; reps 3+4 for pose 1 look generally similar to the traces for pose 2 in Fig 2c. I would adjust the wording here to indicate that only trace 1+2 are highly fluctuating, or provide additional evidence (e.g., a quantitative metric of fluctuation) that indicates that 3+4 are also highly fluctuating.

We realized the inaccurate description of the data here and have revised the words to indicate that only traces 1 and 2 are highly fluctuating.

5) In 193-207: The structures were not previously described as phosphorylated. It is a little confusing why this is suddenly mentioned here but not earlier. The subsequent comparisons were also a bit confusing, though I think I eventually got it. To me, some of the logic seems inverted, e.g., there was a previous observation that Inh-172 binds better to the open channel (In 209-210), therefore it makes sense to investigate whether the binding pocket changes between those states (instead of the current flow, which leads the reader to think there is an Inh-172-induced conformational change, which doesn't make sense and caused me to go in circles for 30 min...)

We deeply apologize for the trouble we caused for the reviewer. We have done the following to amend our mistakes. First, we now have indicated in the text that both the control—E1371Q-CFTR protein alone, and the E1371Q-CFTR/Inh172 are both phosphorylated and ATP-bound. This is why both structures have their NBDs dimerized. Second, we corrected a mistake in line 204 (in the old version file). PDB code 5UAK refers to the unphosphorylated, ATP-free **wild type (WT)** CFTR structure, **NOT** unphosphorylated, ATP-free E1371Q-CFTR structure which is not available as of now. So, when comparing the Inh-172 binding site differences in different states, we refer to the unphosphorylated, ATP-free WT-CFTR (PDB code: 5UAK) as the closed state. It is important to point out that there is currently no open state structure of CFTR available. But the phosphorylated, ATP-bound E1371Q-CFTR represents a state that is closest to the open state because of its dimerized NBDs. Therefore, the two binding sites we are comparing here are from unphosphorylated, ATP-free WT-CFTR (as the closed state) and our own phosphorylated, ATP-bound E1371Q-CFTR/Inh-172. We understand the speculative nature of this comparison. Considering the similar concern from reviewer #2, we now moved this part of the text to the Discussion.

6) Fig. 6: This is an interesting initial finding, that swapping TMD2 between human and pig can more or less change the Inh-172 sensitivity. But I'm not sure it provides much insight into the mechanism, as it is such a large swap.

We were puzzled by the observation that pig CFTR is so insensitive to Inh-172 despite 100% conservation of the residues involved in Inh-172 binding. The higher opening probability with a longer open time of pig CFTR further heightens our confusion since these kinetic properties should favor Inh-172's action (Kopeikin et al., 2010). However, this species difference in Inh-172 sensitivity makes sense once we take the observed conformational changes into consideration. Most notably, our inhibitor-bound structure shows a tilt and lateral motion of the extracellular end of TM8 and TM12 (both in TMD2) with minor changes in the structure of TMD1. The simplest working model for Inh-172 is a two-step reaction for Inh-172-induced inhibition of CFTR function: a binding step followed by a conformational change involving mostly TM8 and TM12. As any motion is relative, movements of TM8 and TM12 must entail forming and breaking of the intimate interactions between these two TMs and others in TMD2. If we assume that the binding step for Inh-172 is the same for pig and human CFTRs given the 100% conservation of the amino acids involved in binding, the hypothesis that differences in TMD2 between these two orthologs may explain their different sensitivity to Inh-172 predicts that

swapping TMD2 should confer this different sensitivity. Indeed, our experiments confirmed this prediction. As a control, now we have carried out experiments where we swapped the whole TMD1 and observed little effect on Inh-172's apparent affinity (see updated **Figure 6**). The results further strengthen the role of TMD2 in determining the sensitivity of CFTR to Inh-172.

7) Throughout this manuscript, the dose response curves are not very well defined. e.g., in Fig 4, the WT curve has no points below ~35%. All the curves in this figure (and many other figures) lack experiments at very low or high concentration, to firmly establish the upper and lower bounds. This could lead to a moderate left-right shift in the curves, depending on the experimental error. I'm not sure it is likely to change any major conclusion, but could certainly change the IC50s. In that regard, the degree of precision for the reported values in Table 1 seems to be a bit much (4 significant figures in some cases). For example, an IC50 of 4023 +/- 281 necessarily has uncertainty in the second digit. I could easily imagine these changing by a factor of 2 if repeated with more points, so again, what is the appropriate degree of precision?

At the lower end of the dose-response curves, the current response in the patch-clamp experiment is extremely slow due to the reduced on-rate of the drug. The conformational change following binding may further contribute to the slow response of CFTR to Inh-172. For instance, at 50 nM of Inh-172, macroscopic WT-CFTR currents already took several minutes to reach a steady-state (**Figure 4a**). It is thus technically challenging to wait for the steady-state current at even lower [Inh-172] (e.g., below 50 nM down to picomolar range). In other words, responses at [Inh-172] lower than 50 nM cannot be assessed accurately.

We agree with the reviewer that the IC50s in Table 1 may not be so precise. Thus, we did not include any digits after the decimal while keeping only the integer part of the values from curve fittings. We did consider changing the unit of the IC50s to μM , or using SI Units, so that the last digit may be rounded, but the effects are not satisfactory: They become less straightforward for comparison. Therefore, we decided to leave the numbers as they are.

Despite an imperfect numerical representation of our results, our conclusion that mutations in the binding site change the apparent affinity of Inh-172 is based on sizable alterations in the response of mutant channels to Inh-172. Most of the differences can be discerned even by eye inspection.

8) Figs S5 and S6 appear to be only cited in Discussion, but introduce new data. They should be cited and discussed in the results, or removed.

We have moved both into the last section of the Results now.

9) Table S1: It is nice to include PDB, EMDB, and EMPIAR codes in the table for easy reference. I would also suggest including map-model CC values (mask, box, peaks, and volume).

We have updated the table with the parameters the reviewer suggested for each dataset.

10) PDB reports indicate 5 residues that don't match the reference sequence of CFTR. What is the source of these additional mutations?

The construct we used includes amino acid deletion and additional mutations. Specifically, they are $\Delta(405-436)$ / M470V/ S492P/ S495P/ A534P/ I539T/ R555K. These mutations help to improve the thermal stability of CFTR proteins (calorimetric $T_m > 70^\circ\text{C}$). It is also important to point out that the stabilized CFTR construct is fully functional, displaying gating properties similar to WT-CFTR (Fig. 6 in Yang et al., 2018). More detailed information on the construct can be found in Yang et al. (2018). Thus, all these changes are with the E1371Q mutant background in our cryo-EM study.

References

- Hwang, T.C., J.T. Yeh, J. Zhang, Y.C. Yu, H.I. Yeh, and S. Destefano. 2018. Structural mechanisms of CFTR function and dysfunction. *J Gen Physiol.* 150:539-570.
- Kopeikin, Z., Y. Sohma, M. Li, and T.C. Hwang. 2010. On the mechanism of CFTR inhibition by a thiazolidinone derivative. *J Gen Physiol.* 136:659-671.
- Linsdell, P. 2014. Cystic fibrosis transmembrane conductance regulator chloride channel blockers: Pharmacological, biophysical and physiological relevance. *World J Biol Chem.* 5:26-39.
- Linsdell, P., and J.W. Hanrahan. 1996. Disulphonic stilbene block of cystic fibrosis transmembrane conductance regulator Cl⁻ channels expressed in a mammalian cell line and its regulation by a critical pore residue. *J Physiol.* 496 (Pt 3):687-693.
- Liu, F., Z. Zhang, A. Levit, J. Levring, K.K. Touhara, B.K. Shoichet, and J. Chen. 2019. Structural identification of a hotspot on CFTR for potentiation. *Science.* 364:1184-1188.
- Liu, X., M. Luo, L. Zhang, W. Ding, Z. Yan, and J.F. Engelhardt. 2007. Bioelectric properties of chloride channels in human, pig, ferret, and mouse airway epithelia. *Am J Respir Cell Mol Biol.* 36:313-323.
- Ma, T., J.R. Thiagarajah, H. Yang, N.D. Sonawane, C. Folli, L.J. Galiotta, and A.S. Verkman. 2002. Thiazolidinone CFTR inhibitor identified by high-throughput screening blocks cholera toxin-induced intestinal fluid secretion. *J Clin Invest.* 110:1651-1658.
- Riordan, J.R., J.M. Rommens, B. Kerem, N. Alon, R. Rozmahel, Z. Grzelczak, J. Zielenski, S. Lok, N. Plavsic, J.L. Chou, and et al. 1989. Identification of the cystic fibrosis gene: cloning and characterization of complementary DNA. *Science.* 245:1066-1073.
- Sheppard, D.N., and K.A. Robinson. 1997. Mechanism of glibenclamide inhibition of cystic fibrosis transmembrane conductance regulator Cl⁻ channels expressed in a murine cell line. *J Physiol.* 503 (Pt 2):333-346.
- Walsh, K.B., K.J. Long, and X. Shen. 1999. Structural and ionic determinants of 5-nitro-2-(3-phenylpropyl-amino)-benzoic acid block of the CFTR chloride channel. *Br J Pharmacol.* 127:369-376.
- Yang, Z., E. Hildebrandt, F. Jiang, A.A. Aleksandrov, N. Khazanov, Q. Zhou, J. An, A.T. Mezzell, B.M. Xavier, H. Ding, J.R. Riordan, H. Senderowitz, J.C. Kappes, C.G. Brouillette, and I.L. Urbatsch. 2018. Structural stability of purified human CFTR is systematically improved by

- mutations in nucleotide binding domain 1. *Biochim Biophys Acta Biomembr.* 1860:1193-1204.
- Yeh, H.I., Y. Sohma, K. Conrath, and T.C. Hwang. 2017. A common mechanism for CFTR potentiators. *J Gen Physiol.* 149:1105-1118.
- Young, P., J. Levring, K. Fiedorczuk, S.C. Blanchard, and J. Chen. 2023. Structural basis for CFTR inhibition by CFTRinh-172. *bioRxiv.2023.2010.2011.561899.*
- Zhang, Z., F. Liu, and J. Chen. 2017. Conformational Changes of CFTR upon Phosphorylation and ATP Binding. *Cell.* 170:483-491 e488.
- Zhou, Z., S. Hu, and T.C. Hwang. 2002. Probing an open CFTR pore with organic anion blockers. *J Gen Physiol.* 120:647-662.

REVIEWER COMMENTS

Reviewer #1 (Remarks to the Author):

No further comments

Reviewer #2 (Remarks to the Author):

The authors have adequately addressed most previous concerns. The following two issues should be addressed:

In their response to major concern #2, the authors write that "the open state of CFTR has a higher affinity for Inh-172 than the closed state". This statement appears to contradict the results in Fig. 8, which show that with ivacaftor (VX-770) the binding affinity to the inhibitor decreased. Please discuss or reconcile this apparent contradiction.

In response to major concern #4, the authors gave the following example of allostery without changes in the ligand binding site: VX-770, which supposedly opens the channel allosterically, doesn't change CFTR structure upon binding. This is not a good example, as VX-770-bound and unbound have identical structures, and hence does not provide evidence for allostery.

Reviewer #3 (Remarks to the Author):

The authors have addressed several of my comments, but a couple remain unresolved and I think should be addressed before publication.

Point 7 (poorly defined dose-response curves): The authors response would seem to explain why points may be missing from the left side of the plot (Fig 4e), but I don't think it explains their absence from the right side. The data points for the three mutants roughly define a straight line. In order to have a meaningful fit that can be used to extract a reliable IC50, one needs to sample higher and lower concentrations than the authors have done. While I understand that there may be technical limitations that make this difficult or impossible, I think scientific rigor still has to be maintained. If the experiment cannot be done properly due to technical limitations, that is fine, but low quality data should not be interpreted and presented. In my opinion, this data is not sufficiently rigorous to analyze and include in a publication. I also still think that the way the authors are reporting significant figures this is not good practice, though this aspect is more of a stylistic/precision issue.

Point 9: I still think depositing the raw micrographs in EMPIAR and including the codes in Table S1 would be ideal. This increases data reusability, transparency, and the development of improved data processing software that benefits the entire structural biology community.

Point 10: I'm not sure if I missed it, but I don't think it is clearly stated in the main text that the work was carried out with an engineered construct; I only see mention of E1371Q and other specific mutations to probe structure/function. In fact, quite the opposite, as the first section title is "Determination of the Structure of Full-length E1371Q-CFTR". Clearly, it cannot be described as full-length if it has a deletion mutation. The authors should include at least 1 sentence in the main text before the first experiment describing the construct they are using. They can either list all the mutations there in the main text or summarizing there that it has 1 deletion plus 6 other mutations and refer the reader to the methods where more details are given directly in the "Cell culture and protein purification" section. There reader shouldn't have to turn to references to know what the authors did in this paper. As a final clarification, when the authors refer to "WT" in the

electrophysiology experiments, are they again referring to this non-WT-engineered construct? Or are those proteins truly WT? If they are mutants as well, that should be clearly stated.

We thank reviewers #2 and #3 for their additional comments that further improve the quality of our manuscript. The response to each comment is detailed below in blue. The corresponding changes in the main text are in red.

Reviewer #2 (Remarks to the Author):

The authors have adequately addressed most previous concerns. The following two issues should be addressed:

In their response to major concern #2, the authors write that "the open state of CFTR has a higher affinity for Inh-172 than the closed state". This statement appears to contradict the results in Fig. 8, which show that with ivacaftor (VX-770) the binding affinity to the inhibitor decreased. Please discuss or reconcile this apparent contradiction.

We apologize for causing such confusion in the revised manuscript for the reviewer. CFTR gating entails a series of complicated conformational changes in both its nucleotide-binding domains (NBDs) and transmembrane domains (TMDs). Previously, we showed that Inh-172 inhibits CFTR gating by a mechanism that does not involve disruption of CFTR's dimerized NBDs, and stabilizing NBD dimer with ATP or phosphate analogs can greatly increase the apparent affinity of Inh-172 (Kopeikin et al., 2010). As NBD dimerization is associated with channel opening, we thus proposed that Inh-172's action is state-dependent: the longer the channel opening, the more stable the inhibitor binding.

Strictly speaking, what Kopeikin et al. (2010) demonstrated is the relationship between Inh-172's apparent affinity and the stability of NBD dimer. This previous work does not exclude the possibility that stabilizing the open state by other maneuvers (e.g., CFTR potentiators such as VX-770) may exert different effects on the apparent affinity of Inh-172. In the current study, we found that VX-770, which does stabilize the open state by acting on CFTR's TMDs, shifts the dose-response relationship of Inh-172 to the right (Fig. 8). Observing that VX-770 and Inh-172 bind to the opposite side of TM8, whose movement is critical for CFTR's opening and closing, we proposed a physical mechanism that VX-770 and Inh-172 counteract with each other on the conformations of TM8 (i.e., a competitive antagonism). Because of this antagonism from VX-770, the potency of Inh-172 is decreased, rather than increased by VX-770. Now we have included more elaboration in the second to the last paragraph in the **Discussion**.

In response to major concern #4, the authors gave the following example of allostery without changes in the ligand binding site: VX-770, which supposedly opens the channel allosterically, doesn't change CFTR structure upon binding. This is not a good example, as

VX-770-bound and unbound have identical structures, and hence does not provide evidence for allostery.

The reviewer is correct that since VX-770 bound and unbound structures are indistinguishable in the binding site, the structural data are not evidence for allostery for CFTR. We nonetheless consider VX-770 (or GLPG1837) to be an allosteric modulator for the following reasons: First, the word allostery does not absolutely require matching changes of binding and gating energetics. In other words, the latter is not a necessity for the former. Thus, no change in the pre- and post-binding structures does not preclude an allosteric modulation mechanism. Indeed, based on the definition of allostery in Brody's Human Pharmacology, VX-770 is an allosteric modulator since it binds to a different location than CFTR's physiological ligand ATP. Second, even when we consider VX-770 itself as a ligand, the fact that its binding site is far away from CFTR's gate again qualifies VX-770 as an allosteric gating modulator. Third, our recent work showed an inverse relationship between the P_o and EC_{50} of GLPG1837—a telltale sign of state-dependent binding of GLPG1837 to CFTR (Yeh et al., 2019). A similar reasoning can be applied to Inh-172. Although Inh-172 bound and Inh-172 free structures have the same binding site structures, Inh-172 does bind to a different location than CFTR's ligand ATP, and its binding also induces conformational changes distant to (i.e. allosteric) the binding site. Hence, Inh-172 should qualify as an allosteric modulator, just like VX-770 and GLPG1837. A failure to observe structural changes in the binding site after ligand binding could well be due to the limited resolution of the cryo-EM structures as described in the **Discussion** section.

We realized that without observing conformational changes at the Inh-172's binding site, this allosteric mechanism deviates from the *classic* allosteric mechanism, which demands a structural change in the binding site induced by drug binding. Therefore, in the last revised version, we cautioned our readers about this deviation in the first paragraph on page 16. Moreover, in the following paragraph, we elaborated further that this could result from imperfect amino acid side chain assignment due to the current resolution limitation. We also specifically pointed out that better resolution of cryo-EM maps in the future may be able to show the fine differences between bound and unbound structures at Inh-172's binding site.

Reviewer #3 (Remarks to the Author):

The authors have addressed several of my comments, but a couple remain unresolved and I think should be addressed before publication.

Point 7 (poorly defined dose-response curves): The authors response would seem to explain why points may be missing from the left side of the plot (Fig 4e), but I don't think it explains their absence from the right side. The data points for the three mutants roughly define a straight line. In order to have a meaningful fit that can be used to extract a reliable IC₅₀, one needs to sample higher and lower concentrations than the authors have done. While I understand that there may be technical limitations that make this difficult or impossible, I think scientific rigor still has to be maintained. If the experiment cannot be done properly due to technical limitations, that is fine, but low quality data should not be interpreted and presented. In my opinion, this data is not sufficiently rigorous to analyze and include in a publication. I also still think that the way the authors are reporting significant figures this is not good practice, though this aspect is more of a stylistic/precision issue.

In Fig. 4e, we added 25 nM and 10 μM data points for WT-CFTR, and the newly added data did not significantly change the shape of the dose-response curve or the fitted IC₅₀. We also added 10 μM data point for M348K-CFTR, so that for all tested mutants, the maximum [Inh-172] is 10 μM (All related figures in the main text are updated). We could not increase [Inh-172] further because precipitates were seen gushing out of our perfusion tubing at concentrations ≥ 20 μM. (See pictures below for the visible precipitates at the bottom of the beaker containing our perfusion solution with the marked concentration of Inh-172.) This technical difficulty is likely encountered by others as well, as Young et al. (2024) also kept the upper bound of their dose-response curves at 10 μM. Nevertheless, the reduced Inh-172 affinity in the tested binding-site mutants is already evident in the raw recording traces, and the rightward shift of IC₅₀ is clear. The main conclusion that mutations in the binding site reduce the apparent affinity of Inh-172 in our electrophysiological experiment should be valid.

Regarding “The data points for the three mutants roughly define a straight line,” this misperception is due to the logarithmic scale of the X-axis. When the X-axis scale is linear, the data points in our original Fig. 4e nicely define a saturation function (see figure below).

For the issues in significant figures, we thank the reviewer’s suggestion and now have changed the unit of IC_{50} from nM to μ M; the reported digits have accordingly been reduced throughout the text and Table 1.

Point 9: I still think depositing the raw micrographs in EMPIAR and including the codes in Table S1 would be ideal. This increases data reusability, transparency, and the development of improved data processing software that benefits the entire structural biology community.

Following the reviewer’s instruction, we have deposited the raw micrographs into EMPIAR for both datasets. The accession codes have been included in Table S1 now.

Point 10: I’m not sure if I missed it, but I don’t think it is clearly stated in the main text that the work was carried out with an engineered construct; I only see mention of E1371Q and other specific mutations to probe structure/function. In fact, quite the opposite, as the first section title is “Determination of the Structure of Full-length E1371Q-CFTR”. Clearly, it cannot be described as full-length if it has a deletion mutation. The authors should include at least 1 sentence in the main text before the first experiment describing the construct they are using. They can either list all the mutations there in the main text or summarizing there that it has 1 deletion plus 6 other mutations and refer the reader to the methods where more details are given directly in the “Cell culture and protein purification” section. There reader shouldn’t have to turn to references to know what the authors did in this paper. As a final clarification, when the authors refer to “WT” in the electrophysiology

experiments, are they again referring to this non-WT-engineered construct? Or are those proteins truly WT? If they are mutants as well, that should be clearly stated.

We thank the reviewer's suggestion and now we have stated the deletion and mutations in our construct in the "Cell culture and protein purification" section. We have also added one sentence in the first paragraph of the **Result** to remind the readers of these details. In addition, we removed "Full Length" from the subtitle of the first section of the **Result** as pointed out by the reviewer. For our electrophysiology experiments, WT construct refers to the native wild-type CFTR construct without any deletion or mutation.

References

- Kopeikin, Z., Y. Sohma, M. Li, and T.C. Hwang. 2010. On the mechanism of CFTR inhibition by a thiazolidinone derivative. *J Gen Physiol.* 136:659-671.
- Yeh, H.I., L. Qiu, Y. Sohma, K. Conrath, X. Zou, and T.C. Hwang. 2019. Identifying the molecular target sites for CFTR potentiators GLPG1837 and VX-770. *J Gen Physiol.* 151:912-928.
- Young, P. G., Levring, J., Fiedorczuk, K., Blanchard, S. C. & Chen, J. Structural basis for CFTR inhibition by CFTR(inh)-172. *Proc Natl Acad Sci U S A* **121**, e2316675121, doi:10.1073/pnas.2316675121 (2024).

REVIEWERS' COMMENTS

Reviewer #3 (Remarks to the Author):

Unless the authors can present a compelling reason to not do so, they should clearly state that they are using an engineered construct in the main text. Clearly, the engineered mutations affect the properties of the protein, which is why it is being used instead of the WT sequence. This should be disclosed and be very easy to fix, and should not require 3 rounds of review to correct. The details can be in the Methods, but as currently written, a reader could reasonably be misled to assume that the construct is WT apart from the E1371Q mutation and perhaps a tag.

We thank the reviewer for the additional comment. Below is our response in blue. The corresponding changes in the main text are red in the first paragraph of the 'Results'.

Reviewer 3: Unless the authors can present a compelling reason to not do so, they should clearly state that they are using an engineered construct in the main text. Clearly, the engineered mutations affect the properties of the protein, which is why it is being used instead of the WT sequence. This should be disclosed and be very easy to fix, and should not require 3 rounds of review to correct. The details can be in the Methods, but as currently written, a reader could reasonably be misled to assume that the construct is WT apart from the E1371Q mutation and perhaps a tag.

We have now revised the first paragraph of 'Results' to ensure that the readers clearly understand that the construct we use is an engineered one.